# Discovery and functional assessment of a novel adipocyte population driven by intracellular Wnt/β-catenin signaling in mammals

**Zhi Liu[1,2†], Tian Chen[1,2†], Sicheng Zhang[1,2], Tianfang Yang[1], Yun Gong[3], Hong-Wen Deng[3], Ding Bai[2], Weidong Tian[2]\*, YiPing Chen[1]\***

[1]Department of Cell and Molecular Biology, Tulane University, New Orleans, United States; [2]State Key Laboratory of Oral Diseases, National Clinical Research Center for Oral Disease, West China Hospital of Stomatology, Sichuan University, Chengdu, China; [3]Tulane Center of Biomedical Informatics and Genomic, Deming Department of Medicine, School of Medicine, Tulane University, New Orleans, United States

**Abstract** Wnt/β-catenin signaling has been well established as a potent inhibitor of adipogenesis. Here, we identified a population of adipocytes that exhibit persistent activity of Wnt/β-catenin signaling, as revealed by the Tcf/Lef-GFP reporter allele, in embryonic and adult mouse fat depots, named as Wnt[+] adipocytes. We showed that this β-catenin-mediated signaling activation in these cells is Wnt ligand- and receptor-independent but relies on AKT/mTOR pathway and is essential for cell survival. Such adipocytes are distinct from classical ones in transcriptomic and genomic signatures and can be induced from various sources of mesenchymal stromal cells including human cells. Genetic lineage-tracing and targeted cell ablation studies revealed that these adipocytes convert into beige adipocytes directly and are also required for beige fat recruitment under thermal challenge, demonstrating both cell autonomous and non-cell autonomous roles in adaptive thermogenesis. Furthermore, mice bearing targeted ablation of these adipocytes exhibited glucose intolerance, while mice receiving exogenously supplied such cells manifested enhanced glucose utilization. Our studies uncover a unique adipocyte population in regulating beiging in adipose tissues and systemic glucose homeostasis.

**\*For correspondence:**
drtwd@sina.com (WT);
ychen@tulane.edu (YPingC)

[†]These authors contributed equally to this work

**Competing interest:** The authors declare that no competing interests exist.

## Editor's evaluation

It is becoming increasingly clear that adipocytes are not homogenous, but rather comprise several distinct subtypes with specific physiologic functions. This work presents evidence for an unexpected subpopulation of adipocytes displaying atypical Wnt signaling. The data suggest a role of these adipocytes in thermogenesis, which could have importance for understanding energy homeostasis.

## Introduction

Adipose tissues, consisting of white adipose tissue (WAT) and brown adipose tissue (BAT), play a critical role in maintaining whole-body metabolic homeostasis, with WAT serving for energy storage and BAT for energy dissipation to produce heat (*Cannon and Nedergaard, 2004*; *Rosen and Spiegelman, 2014*). In addition to white and brown fat cells that are differentiated from heterogenous stromal vascular fractions (SVFs) of distinct lineages (*Hepler et al., 2017*; *Rosenwald and Wolfrum, 2014*; *Schwalie et al., 2018*), a third type of inducible adipocyte exists known as beige or brite adipocytes

that are transiently generated in WAT depots in response to external stimulations such as environmental cold acclimation (*Boström et al., 2012*; *Ikeda et al., 2018*; *Wu et al., 2012*). Like brown adipocytes, activated beige adipocytes through "beiging" or "browning" process of WAT express key thermogenic marker uncoupling protein 1 (UCP1) and exert adaptive thermogenic function (*Ishibashi and Seale, 2010*; *Petrovic et al., 2010*). Despite that the capacity for thermogenic fat cells (i.e. brown and beige adipocytes) to protect against diet-induced obesity and metabolic disorders has been recognized (*Bartelt and Heeren, 2014*; *Crane et al., 2015*; *Hasegawa et al., 2018*; *Ikeda et al., 2017*; *Lowell et al., 1993*; *Tseng et al., 2010*), a comprehensive understanding of developmental origins and regulatory mechanism of beiging is still missing, partially due to the cell-type complexity in adipose tissues. It has been appreciated that beige adipocytes arise by both de novo adipogenic differentiation from progenitor cells and direct conversion/transdifferentiation from existing white adipocytes (*Barbatelli et al., 2010*; *Shao et al., 2019*; *Wang et al., 2013*). While previous profiling studies have identified some specific markers (for example, CD137 and CD81) for progenitors of beige adipocytes (*Oguri et al., 2020*; *Wu et al., 2012*), the cellular origin of beige adipocytes derived from direct conversion remained elusive.

Wnt/β-catenin (canonical) signaling pathway plays a fundamental role in cell proliferation, differentiation, and tissue homeostasis. In the presence of Wnts, 'destruction complex' including glycogen synthase kinase 3 (GSK-3) is inhibited and cytoplasmic β-catenin translocates into the nucleus and interacts with TCF/LEF transcription factors, which is the hallmark of the canonical Wnt signaling activation, to activate downstream target genes (*Cadigan and Liu, 2006*; *Dale, 1998*). In addition, β-catenin nuclear translocation can also be triggered by other intracellular factors such as AKT and Gα$_s$, leading to the activation of β-catenin-mediated signaling in a ligand- and receptor-independent manner (*Fang et al., 2007*; *Regard et al., 2011*). Despite the consensus that Wnt/β-catenin signaling imposes negative effects on adipogenesis by inhibiting Pparγ (*Regard et al., 2011*; *Ross et al., 2000*; *Waki et al., 2007*), clues have been pointing to potential roles of Wnt/β-catenin signaling in adipogenesis and adipose tissue function. For instance, β-catenin and TCF7L2, two key effectors of canonical Wnt signaling pathway, are expressed by mature adipocytes. Adipocyte-specific mutations in *Ctnnb1* (encoding β-catenin), *Tcf7l2*, or *Wls* (encoding Wntless) led to impaired adipogenesis (*Bagchi et al., 2020*; *Chen et al., 2020*; *Chen et al., 2018*). Particularly, loss of *Tcf7l2* in mature adipocytes gave rise to adipocyte hypertrophy, inflammation, as well as systemic glucose intolerance and insulin resistance, implying an important biological role for Wnt/β-catenin signaling in adipose function (*Chen et al., 2018*; *Geoghegan et al., 2019*). However, direct evidence for active Wnt/β-catenin signaling in adipocytes is lacking.

In the current studies, we have revealed an unexpected adipocyte population that is marked by active Wnt/β-catenin signaling intracellularly, named as Wnt⁺ adipocytes. We found that the Wnt⁺ adipocytes are derived from progenitor lineage that is distinct from that of classical adipocytes and emerge at embryonic stage. Using single-cell transcriptomics and chromatin accessibility profiling assays, we showed that these Wnt⁺ adipocytes distinguish from other conventional ones with respect to molecular and genomic signatures and are highlighted by thermogenic properties. Cold exposure studies uncovered that the Wnt⁺ fat cells play a crucial role in thermogenic response via not only undergoing direct conversion to beige adipocytes but also exerting an indispensable role in beige fat recruitment, representing an important type of thermogenesis-regulatory cells. Finally, gain- and loss-of-function studies further revealed the Wnt⁺ adipocytes as a beneficial metabolic determinant in blood glucose control. The fact that this novel population of adipocytes may also exist in humans implicates it as a potential therapeutic target for metabolic diseases.

## Results

### A unique fraction of murine adipocytes displays Wnt/β-catenin signaling activity

In an unrelated study on the effect of the canonical Wnt signaling on osteogenic differentiation of murine bone marrow stromal cells (BMSCs), we used BMSCs from the well-defined Wnt/β-catenin signaling specific reporter mouse line TCF/Lef:H2B-GFP (hereafter T/L-GFP) that allows for real-time monitoring of Wnt/β-catenin activity at single-cell resolution (*Ferrer-Vaquer et al., 2010*). Adipogenic differentiation of T/L-GFP BMSCs was conducted in parallel as a negative control for the activation of

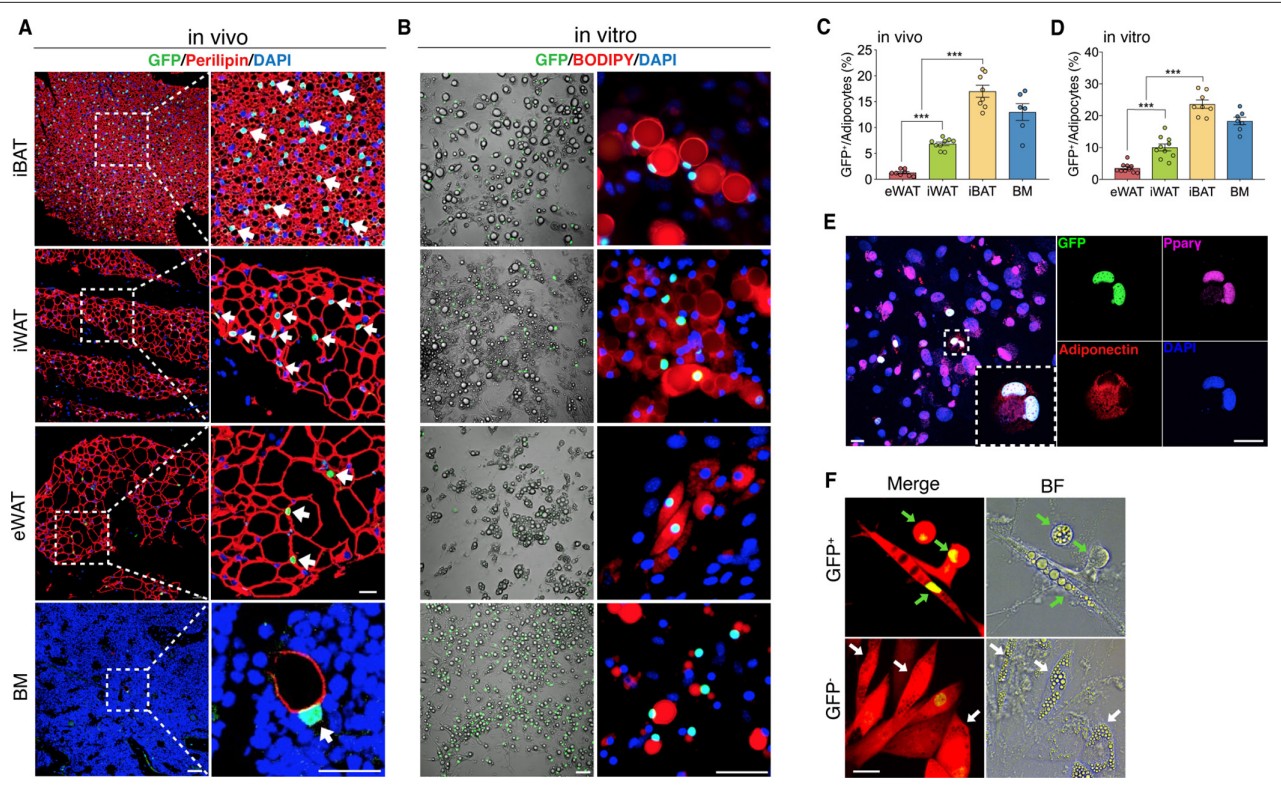

**Figure 1.** The existence of adipocytes exhibiting active Wnt/β-catenin signaling. (**A and B**) Immunofluorescent and microscopy images of active Wnt/β-catenin in vivo (**A**) and in vitro (**B**), indicated by GFP expression from the TCF/Lef:H2B-GFP allele, in adipocytes marked by Perilipin or BODIPY (red). Nuclei were stained with DAPI (blue). Scale bars, 50 μm; close-up scale bars, 20 μm. (**C**) Quantification of Wnt+ adipocytes among total adipocytes in various fat depots of adult male mice in (**A**). n = 6–9 mice. (**D**) Quantification of Wnt+ adipocytes among total adipocytes induced from SVF cells derived from adult male fat depots and BM stroma in (**B**). . n = 7–10 independent experiments. (**E**) Immunofluorescent staining of Ppary (purple) and Adiponectin (red) in cultured GFP+ adipocytes derived from bone marrow in (**B**). n = 3 independent experiments. Scale bars, 20 μm. (**F**) Representative images of GFP+ (green arrows) and GFP- (white arrows) adipocytes induced from human bone marrow stromal cells infected with TCF/Lef:H2B-GFP reporter lentiviral virus. n = 2 independent experiments, 3 independent wells each. Scale bar, 20 μm. Data are mean ± s.e.m., ***p < 0.001, one-way ANOVA followed by Tukey's test.

The online version of this article includes the following figure supplement(s) for figure 1:

**Figure supplement 1.** An inducible mouse model validates the existence of Wnt/β-catenin-positive adipocytes.

**Figure supplement 2.** Age- and sex-dependent dynamics of Wnt+ adipocytes.

**Figure supplement 3.** Infection of lentiviral virus in hBMSCs study.

Wnt/β-catenin signaling. Surprisingly, we observed repeatedly (n > 10) the presence of a small population of GFP-positive cells containing lipid droplets (data not shown). This unexpected observation prompted us to conduct a careful survey for the presence of active Wnt/β-catenin signaling in any adipocytes of fat depots from T/L-GFP mice. In addition to those tissues and cells such as muscle and dermal cells known to manifest active Wnt/β-catenin signaling (*Ferrer-Vaquer et al., 2010*), we virtually observed a subset of Perilipin+ adipocytes that exhibited Wnt/β-catenin signaling activity interspersed within various fat depots, including interscapular BAT (iBAT), inguinal WAT (iWAT), epididymal WAT (eWAT), and bone marrow (BM) in adult mice (*Figure 1A*). Since adipose tissues possess multiple types of non-adipocytes (*Cinti, 2005*), and the nuclei are often squeezed at the edge of mature white adipocytes, it is difficult to determine the exact hosts of the GFP+ nuclei in white adipose tissues. To confirm that these Wnt/β-catenin-positive cells are indeed adipocytes, we created a Tcf/Lef^CreERT2 transgenic allele in which the CreERT2 cassette was placed under the control of the same regulatory elements as that used in the T/L-GFP allele (*Figure 1—figure supplement 1A*). This Tcf/Lef^CreERT2 allele, upon compounding with *Rosa26R*^mTmG reporter allele followed by tamoxifen administration, indelibly labeled the membrane of a fraction of adipocytes, as seen in freshly isolated cells and in sectioned iWAT and iBAT (*Figure 1—figure supplement 1B, C, E, F*). Moreover, addition of the T/L-GFP allele

to Tcf/Lef^CreERT2^;*Rosa26R*^mTmG^ mice produced adipocytes that were tagged by both membrane-bound and nucleus-localized GFP (*Figure 1—figure supplement 1D*). Therefore, we identified a population of adipocytes that displays active Wnt/β-catenin cascade, and accordingly, we referred to these cells as Wnt⁺ adipocytes.

To chart a census of Wnt⁺ adipocytes within distinct mouse fat depots, we quantified Wnt⁺ fat cells in T/L-GFP mice. We observed that the adipocytes expressing nuclear GFP count for various percentages of total fat cells in different fat depots, with the highest level (17.02% ± 3.06%) in iBAT, the lowest one (1.28% ± 0.56%) in eWAT, and relatively abundant percentage (6.86% ± 0.98%) in iWAT, a representative beiging site (*Cannon and Nedergaard, 2004*), of adult male mice (*Figure 1C*). In addition, Wnt⁺ adipocytes could be detected as early as embryonic day 17.5 (E17.5) (*Figure 1—figure supplement 2A, B*) and the proportions varied in different fat depots over the time course of postnatal stages (*Figure 1—figure supplement 2E, F*). Interestingly, sex and age appeared to have an impact on the amounts of Wnt⁺ adipocytes, as the percentage of Wnt⁺ adipocyte in female fat depots was around half as compared to their male counterparts and was dramatically reduced in aged male mice (*Figure 1—figure supplement 2C-F*).

## Wnt⁺ adipocytes can be derived from multiple cellular sources including human BMSCs

To characterize Wnt⁺ adipocytes, we set out to determine if the resident SVF cells within adipose tissues constitute the source of Wnt⁺ adipocytes. Using standard white fat pro-adipogenic induction protocol, we readily induced Wnt⁺ adipocytes in vitro from SVFs derived from iBAT, iWAT, eWAT, as well as BM stroma, as confirmed by staining of the general adipocyte markers Pparγ and Adiponectin (*Figure 1B and E*). The induced Wnt⁺ adipocytes exhibited similar morphology compared to Wnt⁻ (non-GFP labeled) ones and also made up a very similar percentage of total induced fat cells as they are present in their corresponding fat depot (*Figure 1D*), suggesting that the Wnt⁺ adipocytes we observed in vivo are derived from the adipogenic precursors among SVFs. Given the fact that Wnt⁺ adipocytes are present in developing embryos, we asked if embryonic mesenchymal stem cells can differentiate to Wnt⁺ adipocytes as well. Notably, we could induce E13.5 mouse embryonic fibroblasts (MEFs) to differentiate into Wnt⁺ adipocytes in vitro (*Figure 1—figure supplement 2G*), indicating an early developmental origin of Wnt⁺ adipocyte precursors.

To determine if Wnt⁺ adipocytes could also be induced from human stromal cells, we transfected human primary BMSCs with an mCherry-expressing lentiviral vector carrying a TCF/Lef:H2B-GFP reporter cassette (*Figure 1—figure supplement 3A*) and then cultured these cells with pro-differentiation medium. As the positive control, pre-osteocytes/osteocytes differentiated from infected BMSCs under pro-osteogenic conditions exhibited overlapped mCherry expression and nuclear Wnt/β-catenin signaling activity (*Figure 1—figure supplement 3B*), validating the effectiveness of the transduction system. Importantly, under the pro-adipogenic medium, we observed GFP-tagged adipocytes induced from the infected BMSCs (*Figure 1F*), indicating the differentiation of Wnt⁺ adipocytes from human stromal cells. Together, these results suggest that Wnt⁺ fat cells appear to constitute a widespread adipocyte population that originates from embryonic stage and exists in mice and possibly in humans.

## Wnt/β-catenin signaling is activated in Wnt⁺ adipocytes in an intracellular manner

We next addressed how Wnt⁺ adipocytes develop during adipogenesis by conducting real-time monitoring on behaviors of induced Wnt⁺ adipocytes in vitro. No visible GFP expression was found in SVFs in culture prior to pro-adipogenic induction (*Figure 2A*), which is in line with fluorescence-activated cell sorting (FACS) analysis on freshly isolated SVFs (*Figure 2—figure supplement 1A*). Nuclear GFP expression was initially detected in differentiated adipocytes (defined by the presence of lipid droplets) 2 days after adipogenic induction and the GFP signal of the cells being constitutively monitored over the course of adipogenesis was sustained once it was activated (*Figure 2A*), indicating that Wnt/β-catenin signaling does not transiently appear but persists in Wnt⁺ adipocytes. Interestingly, we observed colonized Wnt⁺ and Wnt⁻ adipocytes, respectively, induced from primary BMSCs of T/L-GFP mice after 7-day differentiation, suggesting distinct cell lineages of these two different adipocyte populations (*Figure 2—figure supplement 1B*). This conclusion is supported by the observation that

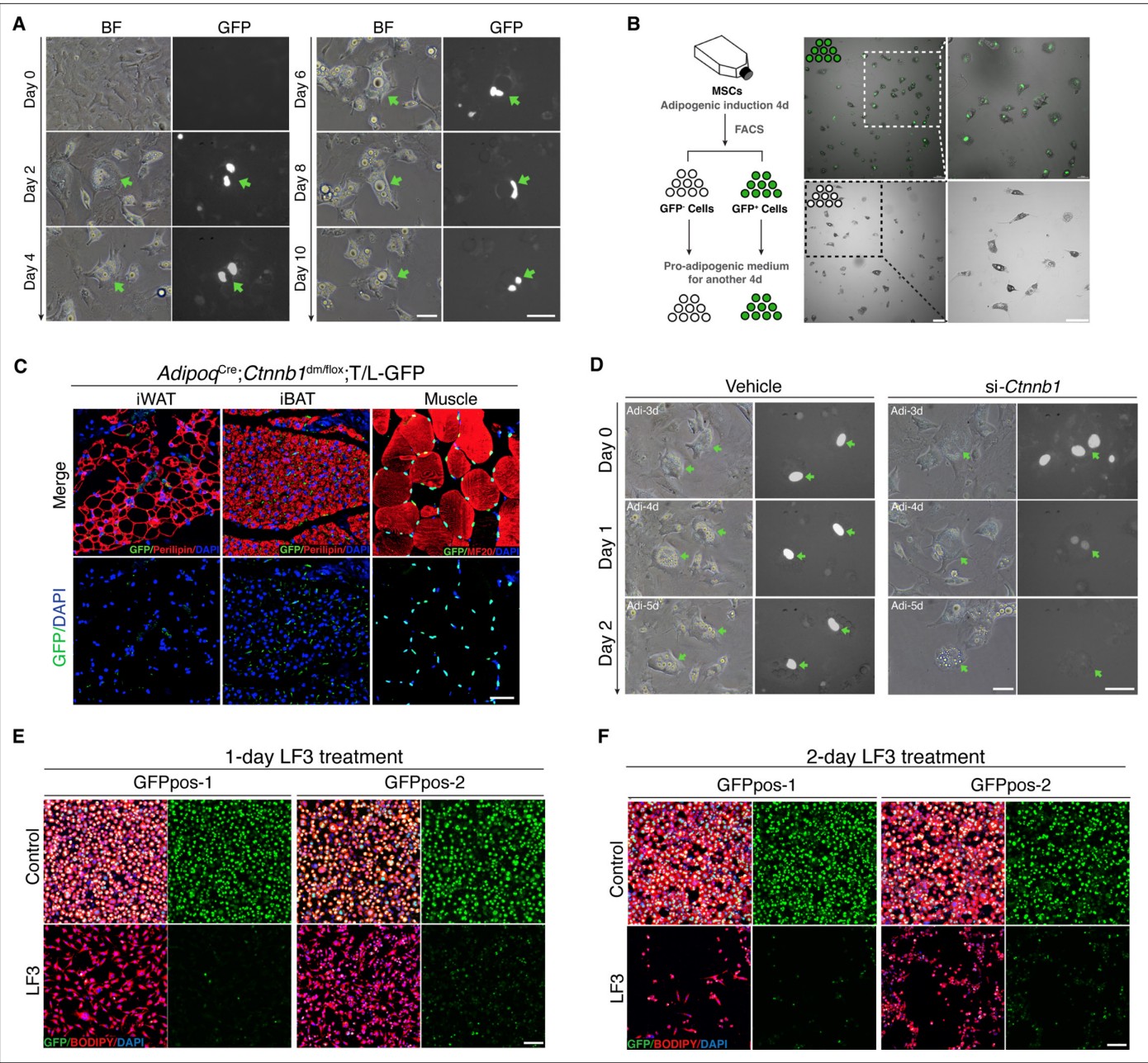

**Figure 2.** Characterization of mouse Wnt+ adipocytes. (**A**) Real-time monitoring microscopy images of Wnt+ adipocytes during adipogenesis, which were induced from BMSCs of T/L-GFP mice. n = 3 independent experiments. Scale bar, 50 µm. (**B**) Schematic of the experiments and microscopy images of separated Wnt+ and Wnt- adipocytes by FACS. n = 3 independent experiments. Scale bars, 100 µm. (**C**) Immunofluorescent images of iWAT and iBAT of *Adipoq*Cre;*Ctnnb1*dm/flox;T/L-GFP mice showing complete absence of Wnt+ adipocytes. Muscle cells were included as positive controls for GFP expression. n = 3 mice. Scale bar, 50 µm. (**D**) Time-lapse microscopy images of induced Wnt+ adipocytes from BMSCs of T/L-GFP mice with *Ctnnb1* and control siRNA-mediated knockdown. n = 2 independent experiments, 3 independent wells each. Scale bars, 50 µm. (**E and F**) Immunofluorescent images of Wnt+ adipocytes induced from two independent cell lines (GFPpos-1 and –2) with LF3 treatment (50 µM) for 1 (**E**) and 2 days (**F**), respectively. LF3 was added into the medium after 3-day pro-adipogenic induction. Note that by 1 day LF3 administration, GFP signals were significantly quenched in Wnt+ adipocytes, along with obviously reduced cell number. By 2-day LF3 administration, remarkable cell death of Wnt+ adipocytes was seen, compared to controls. Scale bar, 100 µm.

The online version of this article includes the following figure supplement(s) for figure 2:

**Figure supplement 1.** Wnt/β-catenin signaling in adipocytes is activated intracellularly.

**Figure supplement 2.** Wnt/β-catenin signaling plays a role in adipogenesis of Wnt- adipocytes.

induced Wnt$^+$ and Wnt$^-$ adipocytes, after separation by FACS and subsequent 4 day in culture, did not convert mutually (**Figure 2B**). The persistent Wnt/β-catenin signaling within Wnt$^+$ adipocyte thus represents an intrinsic cascade activity.

To ensure that the GFP expression in T/L-GFP adipocytes both in vivo and in vitro represents authentic Wnt/β-catenin signaling activity, we first confirmed the presence of active β-catenin and TCF/LEF1 in the nuclei of Wnt$^+$ adipocytes within adipose tissues (**Figure 2—figure supplement 1C, D**). We found that TCF7L2 expression was largely overlapped (~91%) with Wnt$^+$ adipocytes, whereas other TCF proteins (TCF1, TCF3, and LEF1) were barely detected in Wnt$^+$ adipocytes (data not shown). To further confirm that the GFP expression by the T/L-GFP allele is indeed β-catenin dependent, we first set to abolish β-catenin signaling function in vivo. We took advantage of the *Ctnnb1*$^{dm}$ allele that produces truncated β-catenin with abolished transcriptional outputs but retained cell adhesion function (**Valenta et al., 2011**) and the floxed *Ctnnb1* allele by creating mice carrying adipocyte-specific elimination of β-catenin-mediated signaling. Such mice (*Adipoq*$^{Cre}$;*Ctnnb1*$^{dm/flox}$) were crossed to T/L-GFP mice, and the adipose tissues of the resultant mice (*Adipoq*$^{Cre}$;*Ctnnb1*$^{dm/flox-}$;T/L-GFP) were subjected to examination of GFP expression by the T/L-GFP allele. The results showed a complete lack of Wnt$^+$ adipocytes in the absence of β-catenin-mediated signaling in both iBAT and iWAT, indicating the absolute requirement of β-catenin-mediated signaling for the GFP expression in Wnt$^+$ adipocytes (**Figure 2C**). Next, we asked if loss of β-catenin could diminish GFP expression in the Wnt$^+$ adipocytes by conducting in vitro knockdown experiments using short interfering RNA (siRNA) targeting *Ctnnb1* followed by real-time monitoring. *Ctnnb1* knockdown in differentiated Wnt$^+$ adipocytes in which GFP expression had been activated virtually quenched the GFP signals, followed by cell shrinkage (**Figure 2D**). However, surprisingly, we found that DKK1 and IWP-2, both canonical Wnt receptor inhibitors, failed to, but LF3, a molecule that specifically disrupts the interaction between β-catenin and TCF7L2 (**Fang et al., 2016**), did inhibit GFP expression in adipocytes differentiated from T/L-GFP SVFs (**Figure 2—figure supplement 1E-H**). This result indicates that the transcriptional activity of β-catenin-TCF7L2 complex-mediated intracellular Wnt/β-catenin signaling in Wnt$^+$ adipocytes is ligand- and receptor-independent. Remarkably, pharmacological inhibition of Wnt/β-catenin signaling by LF3 also resulted in an overall impaired/delayed adipogenic maturation of SVF-derived adipocytes in a dose-dependent manner (**Figure 2—figure supplement 1E**). Since most of the SVFs in such culture differentiated into Wnt$^-$ adipocytes, this observation suggests the potentially functional importance of Wnt/β-catenin signaling in adipogenesis of Wnt$^-$ adipocytes.

To further define the specific roles of the intracellular Wnt/β-catenin signaling in Wnt$^+$ adipocytes, we immortalized SVF cells (mBaSVF) derived from iBAT of T/L-GFP mouse. By serial limited dilutions, we isolated and established two Wnt$^+$ (GFPpos-1 and GFPpos-2) and two Wnt$^-$ (GFPneg-1 and GFPneg-2, as controls) adipocyte precursor cell lines from mBaSVF cells, respectively. Of note, LF3 treatment of induced Wnt$^+$ adipocytes from both GFPpos-1 and GFPpos-2 cell lines quenched GFP expression in the first day of culture (**Figure 2E**), followed by massive cell death in the second day (**Figure 2F**). By contrast, LF3 administration to induced Wnt$^-$ adipocytes from precursor cell lines did not affect cell viability but slowed down the maturation of Wnt$^-$ adipocytes (**Figure 2—figure supplement 2A-D**), similar to that seen in SVF induced adipocytes (**Figure 2—figure supplement 1E**). However, such LF3-treated Wnt$^-$ adipocytes, once returned to normal pro-adipogenic medium, resumed full adipogenic capacity compared to controls (**Figure 2—figure supplement 2E, F**), indicating that LF3 treatment does not impair but delays the maturation of Wnt$^-$ adipocytes. As controls, the same dose of LF3 showed no impacts on uninduced Wnt$^+$ and Wnt$^-$ precursor cell lines (**Figure 4—figure supplement 1E**).

## Wnt$^+$ adipocytes are diverse from conventional ones with distinct molecular and genomic characteristics

To further distinguish Wnt$^+$ adipocytes from Wnt$^-$ fat cells and explore the global diversity at molecular and genomic levels, we performed single-cell RNA sequencing analysis (scRNA-seq) and single-cell assay for transposase-accessible chromatin sequencing (scATAC-seq) on FACS separated Wnt$^+$ and Wnt$^-$ adipocytes induced from iWAT- and iBAT-derived SVF cells, respectively (**Figure 3A**). After filtering for quality and excluding nonadipocytes (negative for *Adiponectin* expression), bioinformatic analyses classified the input Wnt$^+$ and Wnt$^-$ adipocytes into distinct clusters, indicating two different types of adipocytes (**Figure 3B–E**, **Figure 3—figure supplement 1A-F**). Notably, although the mRNA

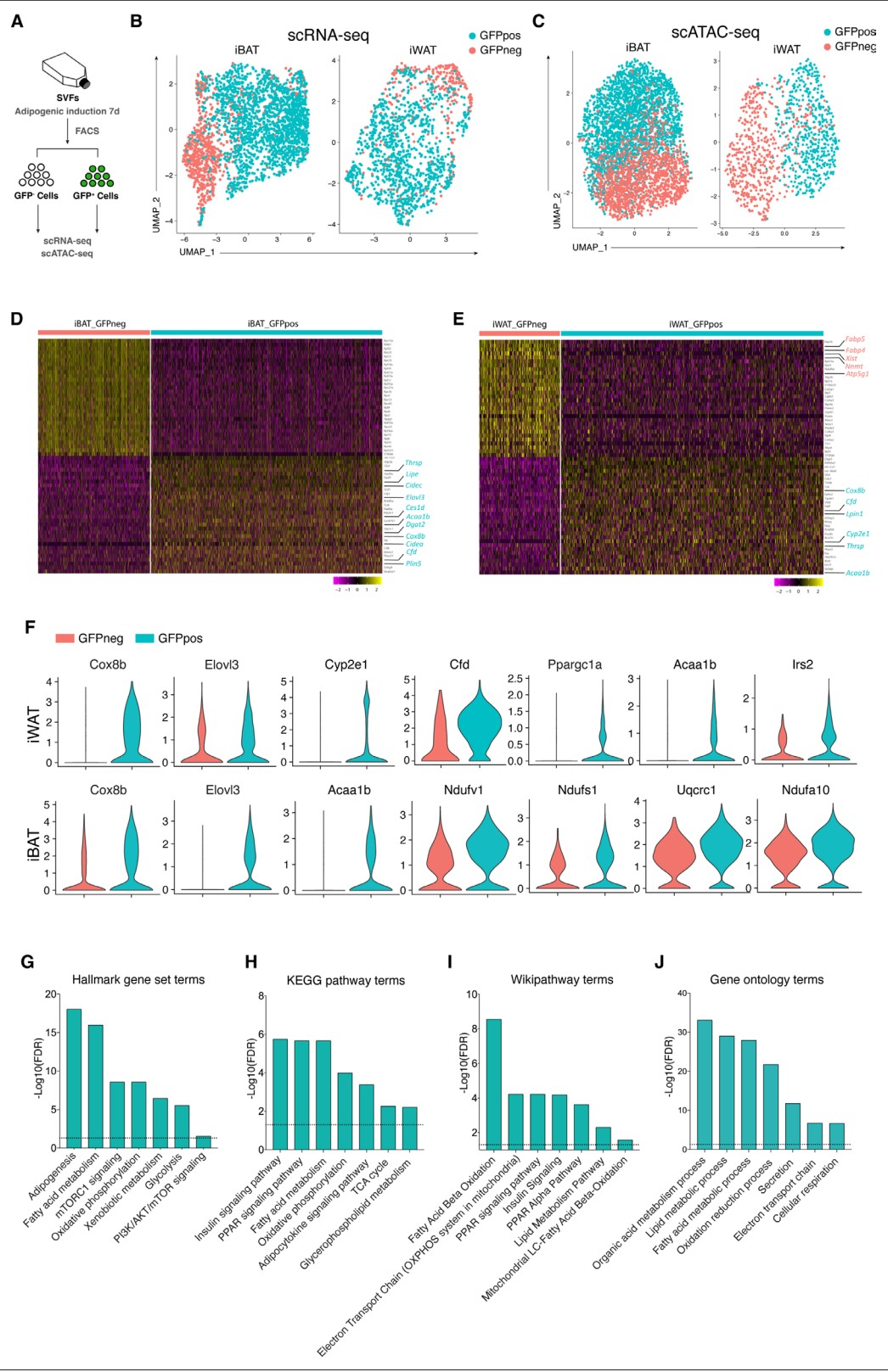

**Figure 3.** Distinct molecular and genomic signatures of Wnt[+] and Wnt[-] adipocytes. (**A**) Schematic of scRNA-seq and scATAC-seq experiments on SVF-induced adipocytes. (**B**) Uniform Manifold Approximation and Projection (UMAP) visualization of 2537 adipocytes from iBAT (1710 GFP[+] and 827 GFP[-]) and 1345 adipocytes from iWAT (984 GFP[+] and 361 GFP[-]) in scRNA-seq. (**C**) UMAP visualization of 4302 adipocytes from iBAT (1727 GFP[+] and 2575 GFP[-])

*Figure 3 continued on next page*

*Figure 3 continued*

and 1383 adipocytes from iWAT (562 GFP$^+$ and 821 GFP$^-$) in scATAC-seq. (**D and E**) Heat maps of expression of top 30 signature genes (***Supplementary file 2***) in iBAT- (**B**) and iWAT-derived (**C**) Wnt$^+$ and Wnt$^-$ adipocytes in scRNA-seq. (**F**) Violin plots of induced Wnt$^+$ and Wnt$^-$ adipocytes showing the distribution of normalized expression values of some representative genes in scRNA-seq. (**G–J**) Hallmark gene sets (**G**), KEGG pathway (**H**), Wikipathway (**I**), and GO Biological Processes ontology (**J**) analyses of DEGs enriched in iWAT-derived Wnt$^+$ adipocytes in scRNA-seq (***Supplementary file 3***).

The online version of this article includes the following figure supplement(s) for figure 3:

**Figure supplement 1.** Wnt$^+$ adipocytes are distinct from Wnt$^-$ ones in molecular signatures.

expression of *Ucp1* was undetectable based on our single-cell transcriptomic data, violin plots showed that several thermogenesis-related genes such as *Cox8b* and *Elovl3* were present at significantly higher levels in Wnt$^+$ adipocytes that were subject to the white fat differentiation condition without browning stimuli (***Figure 3F***), indicative of potentially thermogenic character. Moreover, *Cyp2e1*, a molecule identified as a marker gene in thermogenic regulation (***Sun et al., 2020b***), and *Cidea* were exclusively expressed in a subset of Wnt$^+$ adipocytes induced from iWAT- and iBAT-derived SVFs, as confirmed by immunofluorescent staining (***Figure 3D–F***, ***Figure 3—figure supplement 1G, H***). These facts link Wnt$^+$ adipocytes to thermogenic function in adipose tissues. Enrichment pathway analyses of differentially expressed genes (DEGs) also suggests the primary functions of this population of adipocytes in the regulation of adipogenesis, fatty acid metabolism, and mTORC1 signaling (***Figure 3G–J***). Interestingly, highly enriched expression of *Cfd* that encodes Adipsin, a well-established adipokine that has beneficial impact on maintaining β cell function and plays a key role in systemic glucose homeostasis (***Gómez-Banoy et al., 2019***), was also found in Wnt$^+$ adipocytes (***Figure 3E and F***), implicating a possible role of Wnt$^+$ adipocytes in systemic metabolism. Thus, these results demonstrated that this novel population of intracellular Wnt/β-catenin signaling driven adipocytes is distinct from the classical adipocytes at molecular and genomic levels.

## Insulin/AKT/mTORC1 signaling is required for activation of intracellular Wnt/β-catenin cascade within Wnt$^+$ adipocytes

PI3K/AKT/mTOR signaling modulated by insulin is manifested for promoting cytoplasmic β-catenin accumulation through GSK-3β phosphorylation (***Hermida et al., 2017***; ***Schakman et al., 2008***). This notion upon signaling crosstalk drew our attention because insulin signaling plays pivotal roles in mediating adipogenic differentiation and functionality, and our enrichment pathway analyses also implicate a link to mTORC1 signaling (***Figure 3G–J***). Accordingly, we first assessed AKT signaling activities in induced Wnt$^+$ and Wnt$^-$ adipocytes. Immunoblot assay showed that GFPpos-derived Wnt$^+$ adipocytes exhibited markedly higher levels of AKT phosphorylation than that from GFPneg-derived Wnt$^-$ adipocytes and from mBaSVF-induced adipocytes as unbiased fat cell control (***Figure 4A***). Phosphorylated GSK-3β and 4E-BP1 (Eukaryotic translation initiation factor 4E-binding protein 1), a known substrate of mTOR signaling pathway (***Beretta et al., 1996***), were also found preferentially higher in Wnt$^+$ adipocytes as compared to controls (***Figure 4B***). These results demonstrate enhanced AKT/mTORC1 cascade activity and insulin sensitivity in Wnt$^+$ fat cells.

To test if AKT/mTOR signaling is responsible for intracellular β-catenin-mediated signaling activation in Wnt$^+$ adipocytes, we started with treating iBAT-derived SVF cells with LY294002, a selective PI3K signaling inhibitor, under pro-adipogenic conditions. Again, dramatically reduced number of Wnt$^+$ adipocytes and blunted adipogenesis were observed in a dose-dependent manner (***Figure 4—figure supplement 1A, B***). Consistently, LY294002 administration to GFPpos-induced Wnt$^+$ adipocytes eliminated GFP expression and caused substantial cell death subsequently but did not impact on the survival of GFPneg-derived adipocytes (***Figure 4C and D***, ***Figure 4—figure supplement 1C-E***). To determine a causal role of mTOR signaling in the intracellular activation of β-catenin-mediated signaling in adipocytes in vivo, we subjected T/L-GFP adult mice to an mTOR-specific inhibitor Temsirolimus treatment, which dramatically reduced Wnt$^+$ adipocyte number in both iWAT and iBAT compared to vehicle-injected controls (***Figure 4E and F***). These results demonstrate the dependence of the intracellular β-catenin signaling activity in adipocytes on AKT/mTOR signaling, which is required for cell survival.

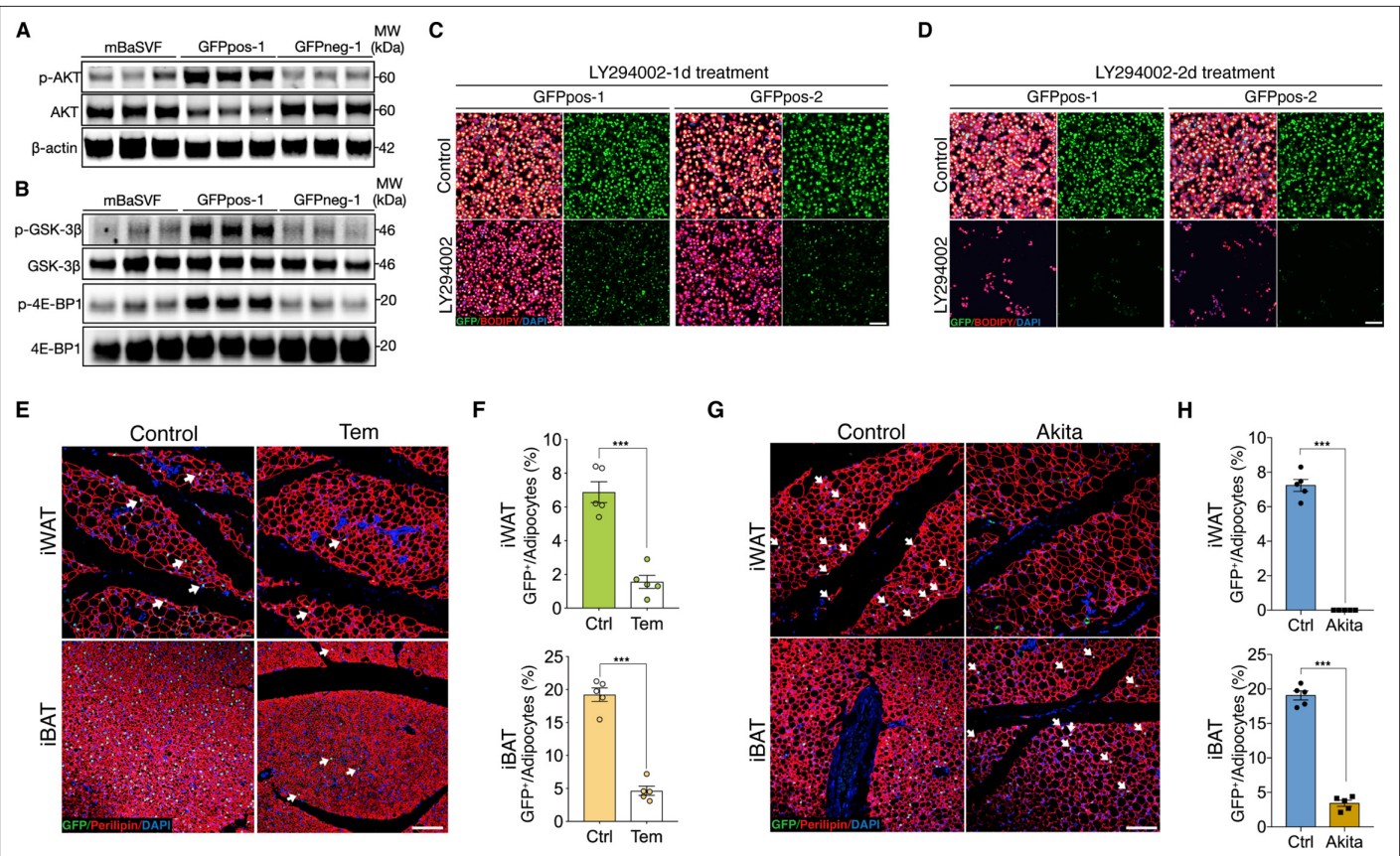

**Figure 4.** AKT/mTOR cascade is indispensable to trigger intracellular β-catenin signaling in Wnt⁺ adipocytes. (**A**) Western blot analysis showing protein levels of AKT, phosphorylated AKT, and β-actin in adipocytes induced from mBaSVF, GFPpos-1, and GFPneg-1 cell lines. n = 3 independent experiments. (**B**) Western blot analysis showing protein levels of GSK-3β, phosphorylated GSK-3β, 4E-BP1, and phosphorylated 4E-BP1 in adipocytes induced from mBaSVF, GFPpos-1, and GFPneg-1 cell lines. n = 3 independent experiments. (**C and D**) Immunofluorescent images of Wnt⁺ and Wnt⁻ adipocytes induced from two independent precursor cell lines (GFPpos-1 and –2) with LY294002 treatment (14 μM) for 1 (**C**) and 2 days (**D**), respectively. LY294002 was added into the medium after 3-day pro-adipogenic induction. LY294002 treatment diminished GFP signals prior to causing marked cell death in Wnt⁺ adipocytes. n = 5 independent experiments. Scale bar, 100 μm. (**E**) Immunofluorescent images of iWAT and iBAT of T/L-GFP mice treated with or without Temsirolimus (Tem). n = 5 mice. Scale bar, 100 μm. (**F**) Quantification of Wnt⁺ adipocytes among total adipocytes in (**E**). (**G**) Immunofluorescent images of iWAT and iBAT of male *Ins2^Akita*;T/L-GFP mice at 8 weeks of age. n = 5 mice. Scale bar, 100 μm. (**H**) Quantification of Wnt⁺ adipocytes among total adipocytes in (**G**). Data are mean ± s.e.m., ***p < 0.001, unpaired Student's t-test.

The online version of this article includes the following source data and figure supplement(s) for figure 4:

**Source data 1.** Full-sized western blot images for *Figure 4A and B*.

**Figure supplement 1.** Inhibition of Akt signaling by LY294002 yields similar results as LF3 treatment.

We next set to explore the downstream targets of the intracellular β-catenin-mediated signaling in Wnt⁺ adipocytes by performing bulk RNA-seq analyses on Wnt⁺ adipocytes with and without LF3 treatment (*Figure 4—figure supplement 1F*). The KEGG pathway analysis of DEGs (fold change >2, FDR < 0.05) showed that the primarily affected (downregulated) pathways in LF3-treated Wnt⁺ adipocytes are 'PI3K/Akt signaling', 'thermogenesis', 'insulin signaling', 'fatty acid metabolism' (*Figure 4— figure supplement 1G*), indicating that the intracellular Wnt signaling indeed mediates the function of the insulin/PI3K/Akt/mTOR pathway.

To further validate the requirement of insulin signaling in Wnt⁺ adipocyte development in vivo, we employed *Ins2^Akita* mice (the so-called Akita mice) in which insulin secretion is profoundly impaired due to genetic defect in the insulin 2 gene (*Yoshioka et al., 1997*) by crossing them to T/L-GFP mice. As expected, Wnt⁺ adipocytes were completely absent in iWAT and substantially reduced in number in iBAT (3.34% ± 0.55%) of 8-week-old male Akita mice (*Figure 4G and H*), demonstrating an indispensable effect of insulin-induced AKT/mTOR signaling on Wnt⁺ adipocytes differentiation and/or maintenance.

# Wnt⁺ adipocytes are required for initiating adaptive thermogenesis in both cell autonomous and non-cell autonomous manners

The potentially highly metabolic and thermogenic characters in Wnt⁺ adipocytes prompted us to investigate the adaptive thermogenic role of Wnt⁺ adipocytes. We started with the examination of mitochondrial activities in SVF-derived adipocytes in vitro and found that Wnt⁺ adipocytes, along with those closely adjacent Wnt⁻ ones, exhibited pronounced lower levels of mitochondrial membrane potential, indicative of higher uncoupling rate, as compared to those Wnt⁻ fat cells located relatively away from Wnt⁺ adipocytes (*Figure 5A and B*). In addition, significantly higher levels of oxygen consumption rate (OCR) were detected in GFPpos-induced Wnt⁺ adipocytes as compared to those GFPneg-derived Wnt⁻ adipocytes and mBaSVF-induced fat cells (*Figure 5C*, *Figure 5—figure supplement 1A, B*), further demonstrating a higher mitochondrial respiration capacity of Wnt⁺ adipocytes.

To explore the possible involvement of Wnt⁺ adipocytes in adaptive thermogenesis in vivo, adult T/L-GFP male mice were subjected to 6 °C temperature for cold challenge. After cold exposure for 2 days to initiate beiging response, we observed the presence of a subset of UCP1-expressing beige adipocytes that were also GFP-positive (*Figure 5—figure supplement 1C*), demonstrating that a portion of beige fat cells arises from Wnt⁺ adipocyte lineage. A close examination revealed a close topological association of UCP1⁺/Wnt⁻ beige adipocytes with UCP1⁺/Wnt⁺ beige cells at this initial beiging stage. Remarkably, as cold exposure was prolonged to 4 days, the number of Wnt⁻ beige adipocytes increased dramatically, but the majority of, if not all, UCP1-labeled Wnt⁻ beige adipocytes were still found present neighboring to Wnt⁺ adipocytes (*Figure 5D*). UCP1⁺/Wnt⁻ beige adipocytes were rarely seen away from Wnt⁺ adipocytes within iWAT, implicating a regulatory role of Wnt⁺ adipocytes in beige fat biogenesis. Similar results were seen in mice treated with β3-adrenergic receptor agonist CL316,243 (*Figure 5—figure supplement 1D*). Importantly, the proportion of Wnt⁺ adipocytes in iWAT was unaltered in response to cold stress or β3-adrenergic receptor agonist treatments, as compared to that at room temperature (*Figure 5—figure supplement 1E*), implying that UCP1⁺/Wnt⁺ adipocytes are converted from existing Wnt⁺ adipocytes but not differentiated from precursor cells. Together with the results from mitochondrial activity assays, these observations suggest a paracrine function of Wnt⁺ adipocytes in modulating beige fat formation.

To determine whether those Wnt⁺ beige adipocytes are converted directly from Wnt⁺ adipocytes under cold conditions, we conducted lineage-tracing studies using Tcf/Lef^CreERT2;*Rosa26R*^mTmG mice. After pre-treatment with tamoxifen and a 2-week washout at ambient temperature, Tcf/Lef^CreERT2;*Rosa26R*^mTmG mice were subjected to cold challenge for 4 days. These mice manifested the presence of mGFP and UCP1 co-labeled adipocytes in iWAT (*Figure 5E*), providing unambiguous evidence for the cell autonomous contribution of Wnt⁺ adipocytes to beiging.

To establish a non-cell autonomous role for Wnt⁺ adipocytes in beige fat biogenesis, we generated a Fabp4-Flex-DTA mouse model (*Figure 5—figure supplement 1F*), a diphtheria-toxin-induced depletion system that allows targeted ablation of Wnt⁺ adipocytes upon crossing to Tcf/Lef^CreERT2 mice (T/L-DTA mice) and tamoxifen administration, which eliminated about 87% of Wnt⁺ adipocytes in iWAT (*Figure 5—figure supplement 1G*). Tamoxifen-treated T/L-DTA mice appeared normal and did not exhibit lipodystrophic phenotype by histological examination under regular chow diet (data not shown). We then challenged Wnt⁺ adipocyte-ablated mice and littermate controls (Fabp4-Flex-DTA mice) at 6 °C temperature. We first measured the core body temperature of each mouse and found that about 31%T/L-DTA mice (5 of 16 mice) developed hypothermia (below 34.5 °C) within 60 hr under cold conditions, with the rest of the T/L-DTA mice exhibiting comparable core body temperature as the controls (*Supplementary file 4*). We next examined if the formation of cold-induced beige fat in iWAT of T/L-DTA mice would be blunted. After 4-day cold stress, DTA-induced ablation of Wnt⁺ adipocytes led to substantially reduced UCP1-expressing beige adipocytes compared to control littermates (*Figure 5F*). This markedly attenuated UCP1 expression persisted even after 2 weeks of cold adaptation (*Figure 5—figure supplement 1H*). The compromised beiging process in iWAT was verified by remarkably decreased expression levels of thermogenic genes assessed by qRT-PCR and western blotting (*Figure 5G and H*). Since the mRNA levels of the pan-adipogenic marker *Adipoq* (*Adiponectin*) appeared similarly between the genotypes (*Figure 5—figure supplement 1I*), it was used as internal reference. As shown in *Figure 5G*, loss of Wnt⁺ adipocytes in T/L-DTA mice caused significantly suppressed expression of thermogenesis-related genes specifically but not those general adipogenic markers in iWAT.

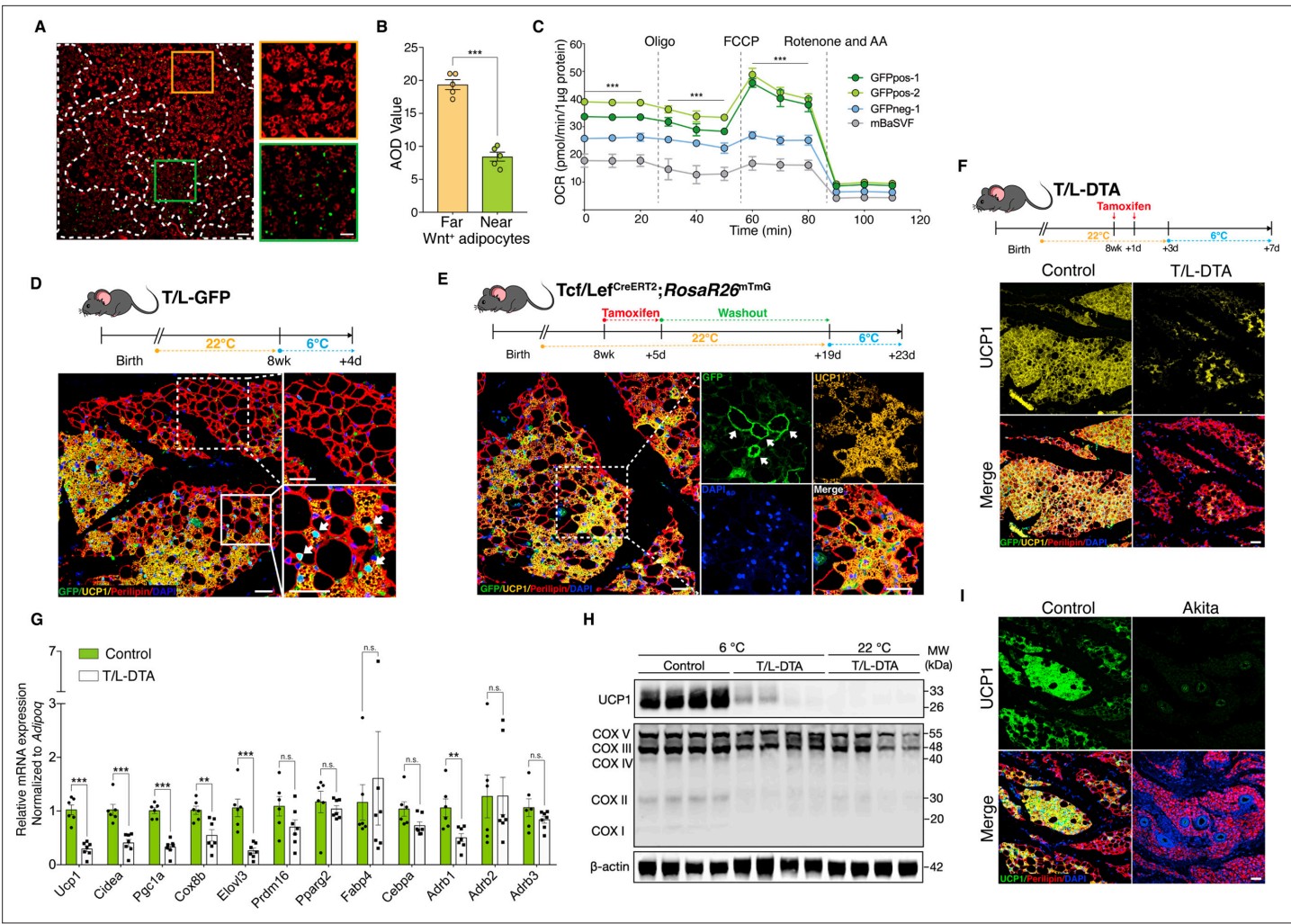

**Figure 5.** Wnt+ adipocytes are essential for initiating adaptive thermogenesis. (**A**) Immunofluorescent staining of mitochondrial membrane potentials in Wnt+ adipocytes induced from iBAT-derived SVF cells of T/L-GFP mice. n = 5 independent experiments. Scale bar, 100 μm; close-up scale bar, 20 μm. (**B**) Quantification of staining in (**A**). AOD, average optical density. Data are mean ± s.e.m., unpaired Student's t-test; ***p < 0.001. (**C**) OCR plots of four groups of adipocytes differentiated from GFPpos-1, GFPpos-2, GFPneg-1, and mBaSVF cell lines, respectively. n = 3 independent experiments. (**D**) Immunofluorescent staining of iWAT from T/L-GFP mice with 4-day thermal challenge showing close association of Wnt+ adipocytes with UCP1+ beige adipocytes. n = 5 mice. Scale bar, 100 μm. (**E**) Immunofluorescent staining of iWAT from tamoxifen-treated Tcf/Lef^CreERT2;Rosa26R^mTmG mice after 4-day cold exposure. n = 8 mice. Scale bar, 50 μm. (**F**) Immunofluorescent staining of iWAT from tamoxifen-treated T/L-DTA mice after 4-day cold exposure. Before cold challenge, mice were rested for 48 hr after final tamoxifen treatment. n = 7 mice. Scale bar, 50 μm. (**G**) Quantitative RT-PCR analysis of gene expression in iWAT from control (Fabp4-Flex-DTA) and T/L-DTA mice in (**F**). n = 6 and 7 mice. Levels of mRNA expression are normalized to that of *Adipoq*. (**H**) Western blot analysis showing protein levels of UCP1, OXPHOS complexes, and β-actin in iWAT from tamoxifen-treated controls and T/L-DTA mice under cold (6 °C) and ambient (22 °C) temperatures. n = 4 mice. (**I**) Immunofluorescent staining for UCP1 (green) and Perilipin (red) in the iWAT from control and Akita mice. Scale bars, 50 μm. Data are mean ± s.e.m., *p < 0.05, ** p < 0.01, ***p < 0.001, n.s., not significant, two-way repeated ANOVA followed by Bonferroni's test (**C**) or unpaired Student's t-test (**G**).

The online version of this article includes the following source data and figure supplement(s) for figure 5:

**Source data 1.** Full-sized western blot images for *Figure 5H*.

**Figure supplement 1.** Wnt+ adipocytes participate in adaptive thermogenesis in iWAT.

**Figure supplement 2.** Two additional mouse models validate the critical role of Wnt+ adipocytes in adaptive thermogenesis in iWAT.

**Figure supplement 3.** Akita mice manifest impaired adaptive thermogenic response.

**Figure supplement 3—source data 1.** Full-sized western blot images for *Figure 5—figure supplement 3B*.

While tamoxifen-induced manipulation of gene expression has been widely utilized in studies of adipose tissue biology, it was suspected that tamoxifen could possibly trigger low level of adipocyte apoptosis and induce non-physiological beiging (*Liu et al., 2015*; *Zhao et al., 2020*). To rule out any contribution of these potential side effects to the deteriorative beiging response in T/L-DTA mice, we further created a Tcf/Lef-rtTA mouse line (*Figure 5—figure supplement 2A-E*) and used it to ablate Wnt⁺ adipocytes upon compounding with TRE-Cre and Fabp4-Flex-DTA mice on doxycycline diet. Such mice (designated as rtTA-DTA) exhibited severely impaired beige fat biogenesis as well upon cold challenge (*Figure 5—figure supplement 2F*). Previous studies have shown that the inactivation of *Pparg* prevents adipogenic differentiation (*Oguri et al., 2020*). In order to exclude the possibility that the inhibited beige fat formation is attributed to potential side effects of DTA-induced cell death on adipose tissues (*Lindhorst et al., 2021*), we further compounded the Tcf/Lef-rtTA allele with TRE-Cre and floxed *Pparg* alleles (*Pparg*$^{F/F}$) to produce TRE-Cre;Tcf/Lef-rtTA;*Pparg*$^{F/F}$ mice (designated as rtTA-*Pparg*$^{F/F}$) in which the differentiation of Wnt⁺ adipocytes was blocked. Identically impaired beiging phenotype was once again observed in such mice after cold stimulation (*Figure 5—figure supplement 2G*). Altogether, these studies support the importance of Wnt⁺ adipocytes in adaptive thermogenesis.

Because Akita mice lack Wnt⁺ adipocytes in iWAT (*Figure 4G and H*), similar to T/L-DTA ones, we tested whether Akita mice also display an impaired adaptive thermogenic capacity by challenging Akita mice at 6 °C. Immunofluorescence staining, qPCR, and WB assays further validated that cold-induced UCP1⁺ cells were largely absent in iWAT of Akita mice (*Figure 5I* and *Figure 5—figure supplement 3A,B*). Collectively, these results demonstrate an essential role of Wnt⁺ adipocytes in initiating beige adipogenesis under cold conditions via non-cell autonomous effect.

## Wnt⁺ adipocytes enhance systemic glucose handling in mice

Given that the presence of Wnt⁺ adipocyte is highly dependent on insulin signaling, we set to evaluate the potential physiological impact of Wnt⁺ adipocytes on whole-body glucose homeostasis. We

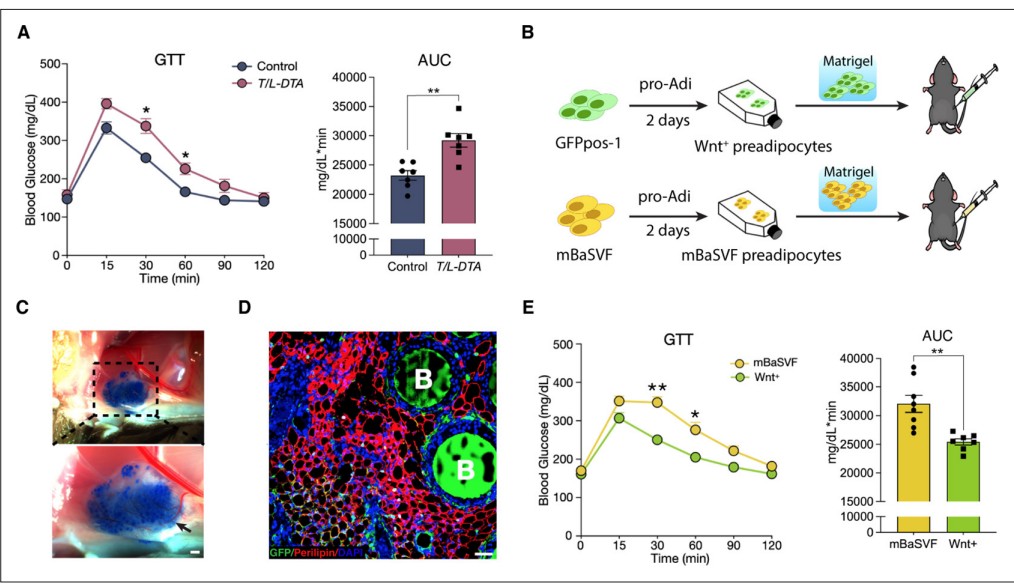

**Figure 6.** Wnt⁺ adipocytes enhance systemic glucose homeostasis. (**A**) Glucose tolerance test (GTT) with calculated area under the curve (AUC) in tamoxifen-treated control (Fabp4-Flex-DTA) and T/L-DTA mice on regular chow diet. n = 7 mice each. (**B**) Schematic of Wnt⁺ adipocyte gain-of-function studies by cell implantation. (**C**) Photographs of fat pad formed by implanted cells. Blue agarose beads were included to locate the Matrigel pad. Black arrow shows benign vascularization of fat pad within two weeks. Scale bar, 50 μm. (**D**) Immunofluorescent staining for Perilipin showing mature adipocytes and accompanied agarose beads (marked by B) in the ectopically formed fat pad in (**C**). Scale bar, 50 μm. (**E**) GTT with calculated AUC in mice that received implantation of committed pre-adipocytes/adipocytes from mBaSVF (n = 8 mice) or GFPpos-1 (n = 7 mice) cell lines for 2 weeks. Data are mean ± s.e.m., *p < 0.05, **p < 0.01, two-way repeated ANOVA followed by Bonferroni's test. AUC was analyzed by two-tailed *t*-test, p = 0.0012 (**A**), 0.0017 (**E**).

started with loss-of-function studies by using T/L-DTA mice, which, under regular chow diet, were subjected to glucose tolerance assays 24 hr after tamoxifen administration. The results showed clearly that T/L-DTA mice displayed impaired glucose handling competence as compared to controls despite similar basal glucose levels (*Figure 6A*), indicating a requirement for Wnt+ adipocytes in handing systemic glucose. We next took gain-of-function approach to test if Wnt+ adipocytes are sufficient to enhance systemic glucose utilization. For this purpose, immortalized GFPpos and mBaSVF cells were induced to become adipogenic commitment after 2 days in pro-adipogenic medium. Committed cells were mixed with Matrigel and implanted into the left abdomen subcutaneous layer of wild-type C57BL/6 J mice under regular chow diet (*Figure 6B*). These implanted cells differentiated into fully mature adipocytes readily and formed vascularized adipose tissues in vivo within 2 weeks (*Figure 6C and D*). Glucose tolerance assays were performed on mice 2 weeks after cell implantation. The results showed that mice bearing implanted Wnt+ adipocytes exhibited a significantly enhanced glucose tolerance as compared to mice receiving mBaSVF-derived adipocytes (*Figure 6E*). Taken together, these results provide appealing evidence for the beneficial impact of Wnt+ adipocytes in systemic glucose homeostasis.

## Discussion

Over the past few decades, numerous studies support the notion that the canonical Wnt signaling pathway functions as a powerful suppressor of adipogenesis (*Longo et al., 2004*; *Ross et al., 2000*; *Waki et al., 2007*). Our current studies expand the conventional dogma conceptually by presenting direct evidence for the existence of a population of adipocytes marked by active intracellular Wnt/β-catenin signaling, revealing the diversity of adipocytes. Despite of a previous study reporting that the canonical Wnt signaling is operative persistently in overall mature adipocytes and plays a role in de novo lipogenesis (*Bagchi et al., 2020*), the activity of such canonical Wnt signaling is apparently below the threshold of the T/L-GFP reporter sensitivity in those so-called Wnt- adipocytes. However, we demonstrated that the β-catenin-mediated signaling in the Wnt+ adipocytes is independent on Wnt ligands and receptors, but is activated intracellularly via signaling crosstalk, in this case, AKT/mTOR signaling. Since β-catenin nuclear translocation and binding to Tcf/Lef1 is regarded as the hallmark of canonical Wnt signaling, we still refer this population of fat cells as Wnt+ adipocytes. Although if AKT/mTOR signaling cascade can activate Wnt/β-catenin signaling remains contradictory (*Baryawno et al., 2010*; *Cabrae et al., 2020*; *Ng et al., 2009*; *Palsgaard et al., 2012*; *Prossomariti et al., 2020*; *Zhang et al., 2013*), it appears to be cell-type dependent. This population of Wnt+ adipocytes was previously neglected, presumably because it is relatively few within fat depots in vivo and most of the prior studies were performed in aggregates of mixed cell types through Wnt ligand- and receptor-dependent manners. Importantly, we showed that the intracellularly AKT/mTOR cascade-dependent β-catenin signaling is required for the survival of Wnt+ adipocytes in vitro. Such intracellular signaling cascade appears to exert distinct function as compared to the ligand-dependent canonical Wnt signaling that regulates lipogenesis in other mature adipocytes (*Bagchi et al., 2020*). This conclusion is supported by the different responses (cell death v.s. recoverable delayed maturation) of Wnt+ and Wnt- adipocytes under the treatment of LF3 that disrupts β-catenin/TCF-mediated signaling specifically in the current studies.

Our single-cell sequencing data showed that Wnt+ adipocytes, as compared to Wnt- adipocytes induced from SVF cells of iWAT and iBAT origins, exhibit distinct signatures at molecular and genomic levels, further supporting the Wnt+ adipocytes as a unique subpopulation of fat cells in mammals. Because FACS-based selection of mature adipocytes presents a huge challenge and the lipid droplets of cultured adipocytes are much smaller, we had to perform sequencing on in vitro cultured cells that were not under their true physiological status. We have attempted to conduct single nucleus RNA sequencing (snRNA-seq) on mature adipocytes isolated from adipose tissues. However, unlike the previous successful studies in which adipocyte nuclei were isolated from adipose tissues simply by one-step enzyme digestion prior to snRNA-seq (*Rajbhandari et al., 2019*; *Sarvari et al., 2021*; *Sun et al., 2020b*), we were not able to obtain desired results due to nuclear membrane breakage after enzyme digestion and following FACS separation of Wnt+ and Wnt- adipocyte nuclei. Nevertheless, under the standard white fat pro-adipogenic induction, SVF-induced Wnt+ adipocytes, but not Wnt- adipocytes, express several brown/beige fat-selective genes, mirroring the thermogenic feature. In addition, pathway analyses indicate highly enriched insulin signaling pathway in Wnt+ adipocytes,

as compared to Wnt⁻ fat cells, consistent with the intracellular activation of β-catenin signaling by AKT/mTOR signaling cascade. A recent study using single-nucleus sequencing (sNuc-seq) revealed multiple unique clusters of adipocytes in mouse and human WAT (*Emont et al., 2022*). By intersecting our scRNA-seq data on Wnt⁺ adipocytes with the published sNuc-seq dataset of mouse iWAT, we found that one cluster (mAd3) of the sNuc-seq dataset, which is relatively enriched in *Tcf7l2*, expresses remarkably high levels of *Cyp2e1* and *Cfd*. These genes, regarded as hallmark of mAd3 cluster, are also highly expressed in Wnt⁺ adipocytes (*Figure 3E and F*). Interestingly, the percentage of mAd3 among the total iWAT adipocytes in chow-fed male group is about 5%, which is very close to that of Wnt⁺ adipocytes in vivo (~7%). Thus, mAd3 possibly represents Wnt⁺ adipocytes in iWAT. Since the activation of Tcf/Lef signaling in the Wnt⁺ adipocytes is relied on Akt/mTOR signaling but not the conventional Wnt ligands and receptors, those traditional downstream markers of Wnt signaling such as *Axins* were not found specifically enriched in Wnt⁺ adipocytes. Therefore, the Akt/mTOR-dependent Wnt signaling in Wnt⁺ adipocytes appears to regulate the expression of genes distinct from that controlled by the conventional Wnt signaling pathway. This conclusion is supported by the results that inhibition of this Akt/mTOR-dependent β-catenin-mediated signaling by LF3 in Wnt⁺ adipocytes negatively impact pathways implicated in 'PI3K/Akt signaling', 'insulin signaling', 'thermogenesis', and 'fatty acid metabolism' (*Figure 4—figure supplement 1G*). While many subpopulations of heterogeneous pre-adipocytes/adipocytes have been identified, the classification of each subset replies mainly on the expression of specific genes such as cell surface markers (*Hepler et al., 2018*; *Oguri et al., 2020*; *Schwalie et al., 2018*; *Wu et al., 2012*). Our current studies, based on the stable feature of an intrinsic cellular signaling identity, provide a new revenue for taxonomy of adipocytes.

In line with their thermogenic feature, as revealed by gene expression profiling, our studies have further uncovered a key role of Wnt⁺ adipocytes in adaptive thermogenesis of adipose tissues. First, Wnt⁺ adipocytes represent at least one of the cell populations that can convert into beige adipocytes directly following cold stress. Interestingly, a conversion of mature white adipocyte to reprogrammed beige characteristics has been postulated to occur predominantly in the 'dormant' or 'latent' beige fat (i.e. thermogenic inactive state of UCP1 lineage adipocytes within WAT depots), which was thought to form initially through de novo beige adipogenesis from progenitors during postnatal development (*Paulo and Wang, 2019*; *Shao et al., 2019*). Because UCP1 expression within iWAT starts at postnatal day 14 independent of environmental temperature (*Wu et al., 2020*), dormant beige adipocytes appear much later in development than Wnt⁺ adipocytes that can be found at embryonic stage. Our recent lineage-tracing studies also revealed non-overlapped Wnt⁺ adipocytes and UCP1⁺ dormant beige cells in iWAT of 4-week-old mice (unpublished data). Therefore, together with the observations that Wnt⁺ and Wnt⁻ adipocytes do not convert mutually in cell culture, our findings demonstrate that Wnt⁺ adipocytes differ from dormant beige cells and represent a population of adipocytes that retain the cellular plasticity and are capable of converting to beige adipocytes directly. Second, Wnt⁺ adipocytes do not just simply function in cell autonomous manner in beiging response, but also exert paracrine effect on and are required for beige fat recruitment. Upon cold exposure, UCP1⁺/Wnt⁻ beige fat emerges largely surrounding Wnt⁺ adipocytes, implicating that Wnt⁺ adipocytes serve as a beiging initiator in a paracrine manner. Remarkably, using targeted cell ablation systems, we showed that depletion of Wnt⁺ fat cells in mice leads to severely impaired beige biogenesis in iWAT in response to cold stress. Similarly impaired beiging response was also seen in iWAT of Akita mice that lack Wnt⁺ adipocytes. These observations underscore the functional importance of Wnt⁺ adipocytes in triggering adaptive thermogenic program of adipose tissue. Despite that the critical factors produced by Wnt⁺ adipocytes to trigger beige adipogenesis remain completely unknown, we conclude that Wnt⁺ adipocytes act as a key regulator of beige biogenesis through both cell autonomous (direct conversion) and non-cell autonomous (recruitment) manners, which is thus far only seen in such adipocyte population. However, it is worth noting that even Wnt⁺ adipocytes exhibit a thermogenic character, whether the intracellular β-catenin signaling plays a direct role in regulating thermogenic function in adipocytes warrants future investigation.

The significant contributions of beige adipocytes to improving whole-body metabolism such as glucose homeostasis and countering diet-induced obesity have been well documented (*Bartelt and Heeren, 2014*; *Crane et al., 2015*; *Harms and Seale, 2013*). GWAS has also linked genetic variations in *TCF7L2*, a key component of canonical Wnt pathway, to type 2 diabetes in humans (*Chen et al., 2018*; *Voight et al., 2010*). We thus speculated that Wnt⁺ adipocytes, driven by insulin/AKT

signaling and functioning as a key regulator of beiging in mice, represent a population of beneficial adipocytes and hold the therapeutic potential for metabolic diseases. Accordingly, our results demonstrate whole-body glucose intolerance in mice with targeted ablation of Wnt+ adipocytes but notably enhanced glucose utilization in mice receiving implantation of exogenously induced Wnt+ adipocytes, highlighting Wnt+ adipocytes as a therapeutically appealing candidate of cell source to restore systemic glucose homeostasis. Consistence with the beneficial function of Wnt+ adipocytes in regulating glucose metabolism is the finding of highly enriched expression of Adipsin coding gene *Cfd* in Wnt+ adipocytes. The evidence that Wnt+ adipocytes could be induced from human primary BMSCs opens a door for future translational research.

## Materials and methods
### Mice
Animal studies were performed according to procedures approved by the Institutional Animal Care and Use Committee of Tulane University. C57BL/6 J wild-type mice (Stock No. 000664), TCF/Lef:H2B-GFP mice (Stock No. 013752), *Rosa26R*$^{mTmG}$ mice (Stock No. 007676), *Ins2*$^{Akita}$ mice (Stock No. 003548), TRE-Cre mice (Stock No. 006234), and *Pparg*$^{F/F}$ mice (Stock No. 004584) were purchased from the Jackson Laboratory and maintained on C57/BL6 background. Mice were raised on a standard rodent chow diet and housed at ambient temperature (22 °C) under a 12 hr light cycles, with free access to food and water. For all experiments, male mice at 8–10 weeks of age were used, unless otherwise stated.

To generate Tcf/Lef$^{CreERT2}$ and Tcf/Lef-rtTA transgenic mice, the coding sequence of the CreERT2 and rtTA (Addgene, 61472) was cloned into pTCF/Lef:H2B-GFP vector by replacing H2B-GFP sequences under the control of the *hsp68* minimal promoter, respectively. Fabp4-Flex-DTA transgenic construct was generated by inserting the coding sequence of diphtheria toxin A (DTA) into the pBS *Fabp4* promoter (5.4 kb) polyA vector (Addgene, 11424), flanked by flip-excision (FLEX) switch. Pronuclear injection and embryo transfer were performed following standard protocols at Tulane Transgenic Animal Facility.

*Adipoq*$^{Cre}$;*Ctnnb1*$^{dm/flox}$;T/L-GFP mice were generated by compounding *Adipoq*$^{Cre}$ (Stock No. 028020; Jackson Laboratory) allele with *Ctnnb1*$^{flox}$ allele (Stock No. 004152; Jackson Laboratory), *Ctnnb1*$^{dm}$ allele (gift from Dr. K. Basler of the University of Zurich), and TCF/Lef:H2B-GFP (T/L-GPF) allele. Adult male mice were subjected to adipose tissue harvest for examination of Wnt+ adipocytes.

For cold-exposure experiments, mice were singly caged and exposed to cold temperature at 6 °C for 2, 4, or 14 consecutive days. For β3-adrenoceptor agonist treatment, adult male mice were injected intraperitoneally with CL316,243 (Sigma-Aldrich, C5976) at 1 mg/kg body weight daily for 4 consecutive days prior to sample collection. Age-matched male littermates were treated intraperitoneally with saline as vehicle controls. For mTOR-specific inhibitor administration, adult male mice were injected intraperitoneally with Temsirolimus (Sigma-Aldrich, PZ0020) at 600 µg/kg (dissolved in 40% ethanol) body weight daily for 5 consecutive days prior to sample harvest. Age-matched male littermates were treated intraperitoneally with 40% ethanol as vehicle controls.

To conduct in vivo lineage-tracing, Tcf/Lef$^{CreERT2}$ allele was compounded with *Rosa26R*$^{mTmG}$ allele to generate Tcf/Lef$^{CreERT2}$;*Rosa26R*$^{mTmG}$ mice that received tamoxifen (dissolved in corn oil) administration intraperitoneally at the dose of 100 mg/kg body weight for 5 consecutive days, and samples were harvested at day 6. For cold exposure study, tamoxifen administrated Tcf/Lef$^{CreERT2}$;*Rosa26R*$^{mTmG}$ mice and littermate controls (Tcf/Lef$^{CreERT2}$ mice that received identical tamoxifen treatment) were rested for tamoxifen washout for 2 weeks before they were housed at 6 °C for 4 consecutive days.

To ablate Wnt+ adipocytes in vivo, Fabp4-Flex-DTA mice were crossed with Tcf/Lef$^{CreERT2}$ mice to generate Tcf/Lef$^{CreERT2}$;Fabp4-Flex-DTA mice (T/L-DTA mice), which received 2-day tamoxifen administration via intraperitoneal injection at the dose of 150 mg/kg body weight. This dose of 2-day tamoxifen administration was found to ablate significantly ablate Wnt+ adipocytes (about 87%) in iWAT of T/L-DTA mice (**Figure 5—figure supplement 1F**). Forty-eight hours later, T/L-DTA mice and tamoxifen-treated littermate controls (Fabp4-Flex-DTA mice) were housed at 6 °C for 2 days, respectively. TRE-Cre;Tcf/Lef-rtTA;Fabp4-Flex-DTA (rtTA-DTA) and TRE-Cre;Tcf/Lef-rtTA;*Pparg*$^{F/F}$ (rtTA-*Pparg*$^{F/F}$) were generated by compounding TRE-Cre and Tcf/Lef-rtTA alleles with Fabp4-Flex-DTA allele or *Pparg*$^{F}$ allele, respectively. For doxycycline administration, chow diet food containing 600 mg/kg doxycycline

(Bio-serv, S4107) (*Zhang et al., 2021*) was given to rtTA-DTA and rtTA-*Pparg*^F/F mice starting from the age of 3 weeks old. Mice were subjected to cold challenge at 8-week-old.

## Isolation of mouse adipose stromal vascular fractions (SVFs) and bone marrow stromal cells (BMSCs)

All cells were isolated from adult TCF/Lef:H2B-GFP male mice, unless otherwise specified. For the isolation of adipose SVF cells, fat tissues were dissected into ice-cold PBS and minced with scissors in a sterile 5 ml tube containing 4 ml of either iBAT digestion buffer (1 X HBSS, 3.5% BSA, and 2 mg/ml collagenase type II) and incubated at 37 °C for 1 hr under agitation or WAT digestion buffer (1 X DPBS, 1% BSA, 2.5 mg/ml dispase and 4 mg/ml collagenase D) for 40 min. The digestion mixture was passed through a 100 μm cell strainer into a 50 ml tube. Digestion was stopped by adding 15 ml PBS containing 2% fetal bovine serum (FBS; Gibco, 10270106) and centrifuged at 500 g for 5 min at room temperature. The supernatant was aspirated, and red blood cells were lysed by incubating the SVF pellet with 2 ml RBC lysis buffer (Invitrogen, 00433357). The number and viability of cells in the suspension were determined using Trypan Blue stain according to the manufacturer's recommendations.

For isolation of BMSCs, femurs and tibias were carefully disassociated from muscle, ligaments, and tendons on ice, and then transferred into a sterile 5-ml tube with DPBS containing 2% FBS. Both ends of the bones were cut to expose the interior of the marrow shaft, and bone marrow was flushed out with MesenCult Expansion medium (STEMCELL, 05513) using a 6-ml syringe and a #23 gauge needle and collected into a 50-ml tube. Cells were gently resuspended and filtered through a 100-mm filter into a collection tube and cell number and viability were determined as described above.

## Fluorescence-activated cell sorting (FACS)

Freshly isolated single-cell suspension of SVFs was diluted to $1 \times 10^7$ cells/ml with FASC sorting buffer (DPBS with 2% FBS, 1 mM EDTA) and the following fluorophore-conjugated antibodies were added: CD31/PE, CD45/PE, and Ter119/PE to enrich lineage⁻ (Lin⁻) populations. Cells were incubated with a cocktail of antibodies on ice for 30 min protected from light and subjected to FACS using a Sony SH800 cell sorter. The cells were selected based on their size and complexity (side and back scatter), and then subjected to doublet discrimination to obtain single-cell signal. The Lin⁻ (CD31⁻ CD45⁻TER119⁻) population was sorted out and plated at the density of $4 \times 10^3$ cells/cm² for further pro-adipogenic induction.

To sort out Wnt⁺ (GFP⁺) and Wnt⁻ (GFP⁻) adipocytes differentiated from the SVFs for scRNA-seq and scATAC-seq, adipocytes were harvested 7 days after pro-adipogenic induction. At this time, the adipocytes became differentiated but contained relatively small size of lipid droplets, which allowed common FACS isolation strategy. 7-AAD was used for viable cell gating. Flow cytometric sorting experiments for adipocytes were performed using FACS-Aria II flow cytometer (BD Biosciences).

## Development of immortalized precursors of Wnt⁺ and Wnt⁻ adipocytes

To establish clonal cell lines of immortalized precursors of Wnt⁺ and Wnt⁻ adipocytes, we generated a lentiviral shuttle vector pUltra-hot-LT that expresses mCherry and Simian Virus 40 Large T antigen (SV40-LT) simultaneously. SV40-LT fragment was amplified from template plasmid pBABE-neo-LargeTcDNA (Addgene, 1780) with primers 5'- gactcatctagagataaagttttaaacagagaggaatctttgcagc –3' and 5'-gcatacggatcctgtttcaggttcagggggagg –3', and the 2145 bp PCR product was subsequently digested with XbaI and BamHI. The vector template pUltra-hot (Addgene, 24130) was linearized with XbaI and BamHI as well. Both the digested PCR product and the linearized pUltra-hot vector were gel-purified and ligated with T4 DNA ligase (Thermo Fisher Scientific, EL0011) according to manufacturer's instructions. The ligated vector is termed pUltra-hot-LT as the final shuttle vector.

For lentivirus (LV) production, 60% confluent monolayers of 293 T cells were transfected with LV shuttle vector pUltra-hot-LT and the packaging plasmids psPAX2 (Addgene, 12260) and pMD2.G (Addgene, 12259) at a molar ratio of 4:3:1. The 293 T cells were cultured in high-glucose Dulbecco's modified Eagle's medium (DMEM; ThermoFisher Scientific, 12430112) with 10% FBS (ThermoFisher Scientific, 16000069), supplemented with 1% Penicillin-Streptomycin (ThermoFisher Scientific, 15140148) and 1% non-essential amino acids (ThermoFisher Scientific, 11140050). The transfections were performed using Helix-IN transfection kit (OZ Biosciences, HX10100) following manufacturer's instructions. Forty-eight hrs after transfection, the 293 T culture supernatants were harvested and

passed through a 0.45 μm pore-sized, 25 mm diameter polyethersulfone syringe filters (Whatman, 6780–2504). LV particles were concentrated from the filtered supernatants using a Lenti-X concentrator kit (Takara, 631232) following manufacturer's protocol.

SVFs of iBAT from adult T/L-GFP male mice were separated and collected as described above. Cells at passage 1 were transduced with pUltra-hot-LT LV particles. Two days after transduction, cells were purified by performing FACS to remove mCherry⁻ and Lin⁺ cells. Immortalized cells were subjected to clonal selection through serial limited dilutions. The expression of GFP in cell colonies under pro-adipogenic induction was used as a readout for the accurate establishment of Wnt⁺ and Wnt⁻ adipocyte cell lines, respectively.

Two precursor cell lines of Wnt⁺ adipocytes, named as GFPpos-1 and GFPpos-2, and two precursor cell lines of Wnt⁻ adipocytes (named as GFPneg-1 and GFPneg-2) were isolated. All of them were confirmed to differentiate into lipid-accumulated adipocytes under pro-adipogenic induction. Both GFPpos-1 and GFPpos-2 precursor cell lines exhibited >95% GFP-positive adipocytes after adipogenic induction, while no cells showed GFP expression in the adipocytes induced from GFPneg-1 and GFPneg-2 cell lines. These cell lines, together with their immortalized parental cells mBaSVF, were tested negative for mycoplasma contamination.

## In vitro pro-adipogenic differentiation of mouse cells

Immortalized cell lines, isolated adipose SVF cells, MEFs, and BMSCs were seeded into plates at the density of $4 \times 10^3$ cells/cm² (cell line and SVF) or $1.5 \times 10^4$ cells/cm² (BMSCs) in MesenCult Expansion medium, and treated with MesenCult Adipogenic Differentiation cocktail (STEMCELL, 05507) with 90% cell confluence. Cells were cultured for various days after pro-adipogenic induction prior to being harvested for following experiments.

Adipogenesis was examined by lipid staining. Briefly, differentiated adipocytes were washed twice with PBS, fixed in 4% paraformaldehyde (PFA) for 15 min, and then stained with Oil Red O solution (Sigma-Aldrich, O1391) for 20 min at ambient temperature or with 20 nM BODIPY in PBS for 15 min at 37 °C. Subsequently, cells were washed with PBS before imaging.

For inhibition studies, SVF cells or precursor cell lines, 3 days after pro-adipogenic induction, were treated with each of the following molecules for 4 days: DKK1 (100 ng/ml), IWP-2 (5 μM), LF3 (10 and 20 μM, or 50 μM on cell lines), and LY294002 (2 and 14 μM), along with vehicles (PBS or DMSO).

## SiRNA-mediated knockdown experiment

To knockdown *Ctnnb1* in adipocytes, siRNA probes were purchased from IDT (TriFECTa DsiRNA Kit) and transfected into BMSCs that had been under pro-adipogenic induction for two days. Cells at a density of $5 \times 10^4$ cells/cm² were plated with 10 nM of a given siRNA dissolved in 1.5% Lipofectamine RNAiMAX (Invitrogen, 13778150) in Opti-MEM I reduced serum medium (Invitrogen, 31985062) and pro-adipogenic differentiation medium (STEMCELL, 05507). The medium was replaced after 24 hr incubation. Time-lapse imaging was carried out to record the real-time change of GFP signal and cell morphology for the next 2 days. At the end, cells were harvested.

## Detection of activated Wnt/β-catenin signaling in adipocytes differentiated from human BMSCs

Primary human bone marrow stromal cells (hBMSCs) from a 29-year-old Caucasian male were purchased from Lifeline Cell Technology (FC-0057 Lot #06333). Lentiviral shuttle vector pUltra-hot-Tcf/Lef:H2B-GFP was generated to serve as Wnt/β-catenin signaling reporter, with mCherry as a reporter for successful transduction of hBMSCs. The inserted fragment Tcf/Lef:H2B-GFP was amplified from a template plasmid (Addgene, 32610) with primers 5'- agagatccagtttggttaattaatattaaccctcactaaagg –3' and 5'- tggagccgacacgggttaatttacttgtacagctcgtc –3'. The 2335 bp PCR product was subsequently gel-purified. The vector template *pUltra-hot* (Addgene, 24130) was linearized with PacI and gel-purified. The purified PCR products and linearized vectors were assembled using a GenBuilder Plus kit (Genscript, L00744-10) according to the manufacturer's instructions. The assembled vector was termed pUltra-hot-Tcf/Lef:H2B-GFP as the final shuttle vector. LV production was described as above.

Human BMSCs were plated at a density of $5.0 \times 10^4$ cells/cm² in Stemlife MSC-BM Bone Marrow Medium (Lifeline Cell Technology, LL-0026). When cultured cells reached 60–70% confluency, concentrated LV pellets were resuspended and mixed with LentiBlast reagent A and B (1:100)

(OZBIOSCIENCES, LBPX500) in a freshly prepared medium and incubated with cells. Forty-eight hrs after viral transduction, hBMSCs were cultured in either pro-adipogenic induction medium (Lifeline Cell Technology, LL-0059) or pro-osteogenic induction cocktail (Lifeline Cell Technology, LM-0023) as positive control for Wnt/β-catenin signaling activity for 7 days, respectively. Cells were subsequently subjected to immunofluorescent assays.

## Immunofluorescence

Adipose tissue was fixed in 4% PFA at 4 °C overnight, followed by dehydration through serial ethanol. Samples were processed into paraffin-embedded serial sections at 6 μm. For immunostaining, paraffin-embedded sections were deparaffinized in xylene and subsequently rehydrated. After the incubation of the slides in boiling Tris-EDTA antigen retrieval buffer for 10 min, the tissues were blocked in PBS containing 10% BSA for 60 min, followed by incubation with primary antibodies against GFP (1:500), Perilipin (1:500), UCP1 (1:300), active β-catenin (1:100), Tcf3 (1:500), and Tcf1 (1:100) at 4 °C overnight. Slides were then incubated with secondary antibodies (1:500) at room temperature for 60 min. After washing, sections were processed with Autofluorescence Quenching Kit treatment (Vector, SP-8400) for 5 min to remove unspecific fluorescence on sections due to red blood cells and structural elements such as collagen and elastin. Sections were stained with 4′,6-diamidino-2-phenylindole (DAPI) and mounted with mounting medium (Vector, Vibrance Antifade). Images of tissue samples were captured using a Nikon confocal Microscope A1 HD25 and analyzed using the ImageJ software (Version 1.51 S). For quantification of Wnt$^+$ adipocytes, we randomly chose 12–18 images from each tissue of one mouse. Microscopic pictures (10 x objective) were taken, and the total cell number and the number of GFP$^+$ adipocytes within each image were counted.

For cryosections, samples were dehydrated in 30% sucrose PBS solution overnight at 4 °C, embedded in optimal cutting temperature compound (Tissue-Plus; Fisher Healthcare), and frozen by liquid nitrogen for solidification. Embedded samples were cryosectioned (Leica, CM1860) at 8 μm and subjected to immunofluorescent staining with primary antibodies against Tcf4 (1:1:00) and Tcf1 (1:100) as described above.

For immunostaining on cell culture, cells were fixed with 4% PFA for 15 min at ambient temperature and then permeabilized in 0.25% Triton X-100 in PBS for 10 min, followed by blocking with 10% BSA in PBS for 30 min. Cells were then incubated with primary antibodies against GFP (1:500), Perilipin (1:500), Runx2 (1:300), mCherry (1:200), Ppary (1:400), Adiponectin (1:100), Cyp2e1 (1:200), and Cidea (1:100) at 4 °C overnight, respectively. Subsequently, cells were stained with secondary antibodies (1:500) at room temperature for 60 min, followed by counterstaining with DAPI. Images were obtained using a Nikon confocal Microscope A1 HD25 and analyzed using the ImageJ software.

## Western blotting

Proteins were extracted from iWAT or cultured adipocytes using Minute Total Protein Extraction Kit (Invent Biotechnologies, AT-022) according to manufacturer's instructions. Twenty μg of proteins were separated by SDS-PAGE (Invitrogen, NW04127BOX) and transferred onto a 0.22 μm Nitrocellulose membrane (LI-COR Biosciences, 926–31092). Membranes were blocked in Tris-buffered saline (TBS) with 0.1% Tween 20% and 5% BSA for 1 hr, followed by overnight incubation with primary antibodies against AKT (1:1,000), p-AKT (1:2,000), GSK-3β (1:1,000), p-GSK-3β (1:500), 4E-BP1 (1:1,000), p-4E-BP1 (1:1,000), UCP1 (1:1,000), OXPHOS cocktail (1:250), or β-actin (1:10,000) in blocking solution at 4 °C overnight. Samples were then incubated with secondary antibodies conjugated to IRDye 800 or IRDye 680 (LI-COR Biosciences) diluted at 1:5000 for 1 hr. Immunoreactive protein was detected by Odyssey Imaging System (LI-COR Biosciences).

## RNA preparation and quantitative RT-PCR

Total RNA was extracted from tissue or cells according to protocol for the RNeasy Lipid Tissue Mini Kit (Qiagen, 74804) or RNeasy Mini Kit (Qiagen, 27106) accordingly. Complementary DNA was synthesized using RevertAid First Strand cDNA Synthesis kit (Thermo Scientific, K1622) according to the protocol provided. Quantitative PCR was performed using a C1000 Touch thermal cycler (BioRad, CFX96 Real-Time System). Values of each gene were normalized to reference genes, *36B4* or *Adipoq*, using the comparative Ct method. Primer sequences are provided in *Supplementary file 1*.

## ScRNA-seq and scATAC-seq

SVF cells isolated from iBAT (3 males and 4 females) and iWAT (10 males and 10 females) of adult T/L--GFP mice were cultured and induced to differentiate into adipocytes as described above. Following separation by FACS, approximately 3000 iBAT-derived GFP$^+$, iBAT-derived GFP$^-$, iWAT-derived GFP$^+$ cells each, and 4000 iWAT-derived GFP$^-$ cells were loaded using Chromium Single Cell 3' v2 Reagent Kit, and libraries were separately prepared and sequenced together on the Illumina NextSeq 550 Sequencing System as 150 bp paired-end reads. Sequencing data were processed on CellRanger v.3.1.0 pipeline with default parameters, including demultiplexing, conversion to FASTQs using bcl2fastq2 v.2.27.1 software, alignment to the mm10 mouse reference genome, filtering, and unique molecular identified (UMI) counting. Gene expression count matrixes were obtained with 2,329 iBAT GFP$^+$ cells (32,539 mean reads and 1313 median genes per cell), 2666 iBAT GFP$^-$ cells (29,534 mean reads and 2076 median genes per cell), 2318 iWAT GFP$^+$ cells (35,693 mean reads and 2282 median genes per cell), and 3137 iWAT GFP$^-$ cells (24,360 mean reads and 2565 median genes per cell). Further scRNA-seq data analysis was performed using the Seurat (*Satija et al., 2015*) package v.3.2.2. Low-quality cells with fewer than 500 detected genes, doublets with more than 4,000 transcripts, and cells with mitochondrial fraction rate higher than 40% were excluded from the analysis. Normalized and scaled data were clustered using the top 20 significant principal components of highly variable genes with the parameter "resolution = 0.7". Uniform Manifold Approximation and Projection (UMAP) dimensionality reduction was performed to visualize the resulting clusters. To filter out non-adipocytes, preadipocytes, and endothelial cells, cell clusters identified by high expression levels of precursor marker *Pdgfra* (*Sun et al., 2020a*) and endothelial marker *Cdh5* (*Corada et al., 2001*) were removed, and *Adipoq*-expressing cells were considered as adipocytes (*Lara-Castro et al., 2007*) that were reperformed to assign clusters (*Figure 3—figure supplement 1A*). The final reported datasets consist of 1,710 Wnt$^+$ and 827 Wnt$^-$ adipocytes for iBAT, 984 Wnt$^+$ and 361 Wnt$^-$ adipocytes for iWAT, respectively (*Figure 3B*). Top marker genes of each cluster and library were determined by the *FindAllMarkers* function with the parameters "only.pos = TRUE, min.pct = 0.2, logfc.threshold = 0.25". Heat maps of top 30 enriched genes from each library was generated using the *DoHeatmap* function (*Figure 3D and E*). Hallmark gene sets, GO Biological Processes ontology, KEGG pathway, and Wikipathway analyses were performed using the Molecular Signatures Database (MSigDB) (https://www.gsea-msigdb.org/gsea/msigdb) to estimate the functional enrichment and biological pathway of input Wnt$^+$ adipocytes based on the differential gene expression (FDR < 0.05) (*Figure 3G–J*).

For scATAC-seq experiment, SVF cells from adult T/L-GFP mice (11 males for iBAT; 7 males and 2 females for iWAT) were induced and adipocytes were acquired as the same workflow for scRNA-seq used (*Figure 3A*). To isolate nuclei, cells were lysed for 4 min on ice according to 10 x genomics protocol (CG000169 Rev D). ScATAC-seq libraries were prepared using the 10 x Genomics platform with the Chromium Single Cell ATAC Library & Gel Bead Kit as recommended by the manufacturer and sequenced on the Illumina NextSeq 550 Sequencing System as 150 bp paired-end reads. Peak matrixes and metadata were generated by Cell Ranger ATAC v.1.1.0 pipeline with default parameters and aligned to the mm10 mouse reference genome. Overall, scATAC-seq datasets containing 3,284 iBAT GFP$^+$ cell (17,706 median fragments per cell), 4650 iBAT GFP$^-$ cells (9,994 median fragments per cell), 766 iWAT GFP$^+$ cells (35,731 median fragments per cell), and 1034 iWAT GFP$^-$ cells (42,083 median fragments per cell) were obtained. Further data analysis was performed using the Signac (*Stuart et al., 2019*) v.1.0.0 package. In brief, outlier cells with <2000 or > 40,000 peaks (iWAT GFP$^+$ > 60,000 peaks, iWAT GFP$^-$ > 90,000 peaks), < 20% reads in peaks, > 0.025 blacklist ratio, > 4 nucleosome signal or TSS-enrichment <2, were considered as low-quality cells or doublets and were removed from downstream analyses. Data of each experiment based on the 95% most common feature were then normalized and scaled through the *FindTopFeatures* and *RunSVD* functions. UMAP and k-nearest neighbor (KNN) were applied to perform non-linear dimension reduction using latent semantic indexing (LSI) (*Cusanovich et al., 2018*) and clustering analysis with the parameter of "resolution = 0.7". To exclude potential non-adipocytes, gene activity matrix for each experiment was generated by summarizing the accessibility in promoter (TSS and up to 2 kb upstream) and cells with high activity of adipocyte specific marker *Adipoq* (*Lara-Castro et al., 2007*) were retained for further analysis. We merged the datasets of Wnt$^+$ and Wnt$^-$ adipocytes derived from iBAT and iWAT, respectively, and applied UMAP for the data visualization (*Figure 3C*, *Figure 3—figure supplement 1B*). Differential chromatin accessibilities between Wnt$^+$ and Wnt$^-$ adipocytes were visualized by the

*CoveragePlot* function (**Figure 3—figure supplement 1C**, D), and motif activities were computed through *Chromvar* (**Schep et al., 2017**) within Signac (**Figure 3—figure supplement 1E**, F).

## Bulk RNA-seq

For bulk RNA-seq assay, Wnt[+] adipocytes from immortalized GFPpos-1 precursor cells, after 3 day pro-adipogenic induction, were subject to LF3 treatment for one day to suppress the intracellular β-catenin-mediated signaling. Wnt[+] adipocytes without LF3-treated were included as control groups. Total RNAs were extracted from control and treatment groups (n = 3 each) as described above and prepared using a NEBNext Ultra II Directional RNA Library Prep Kit for Illumina (NEW ENGLAND BioLabs) according to the manufacturer's instruction. The normalized dsDNA libraries were pooled and sequenced on Illumina NextSeq 1,000 and NextSeq 2000 with P2 (100 cycles) kit at the Center for Translation Research in Infection and Inflammation in Tulane School of Medicine. Sequenced libraries were analyzed as previously described (**Liu et al., 2019**). In brief, HISAT2 (**Pertea et al., 2016**), featureCounts (**Liao et al., 2014**), and DESeq2 (**Love et al., 2014**) softwares were used for reads alignment to mouse reference genome (version mm10), quantification of transcript abundances, and differential gene expression analysis. KEGG pathway analysis of DEGs (fold change >2, FDR < 0.05) was performed by using R package clusterProfiler (**Yu et al., 2012**). The applied DEGs and pathways can be found in **Supplementary file 5**.

## Mitochondrial membrane potential

Mitochondrial membrane potentials of Wnt[+] adipocytes were measured through MitoTracker Deep Red (far red-fluorescent dye) staining. Briefly, SVF cells derived from iBAT were isolated and collected with the method described above. After 7 days of pro-adipogenic induction, cells were incubated with 100 nM MitoTracker Deep Red FM (Invitrogen, M22426) for 30 min at 37 °C. After washing twice with PBS, GFP (abs/em ~ 488/509 nm) fluorescence of differentiated adipocytes and MitoTracker Deep Red (abs/em ~ 644/665 nm) of active mitochondria were monitored using the Nikon confocal Microscope A1 HD25 and analyzed using the ImageJ software.

## Respiration measurements

Cellular oxygen consumption rate (OCR) was measured using the Seahorse XFe24 analyzer (Agilent Technologies). GFPpos-1, GFPpos-2, GFPneg-1, and mBaSVF cells were seeded in an XF24 cell culture microplate (Agilent Technologies, 102342–100) at a density of 20,000 cells per well. After 5 days of pro-adipogenic induction, 500 µl XF assay medium containing 1 mM pyruvate, 2 mM glutamine, and 10 mM glucose was added to each well. Cells were subjected to a mitochondrial stress test by adding oligomycin (5 µM), FCCP (1.25 µM), and rotenone and antimycin A (5 µM) according to the manufacturer's instructions (Agilent Technologies, 103015–100).

## Core body temperature measurement

Core body temperatures of T/L-DTA mice and control littermates were obtained at RT, 6-, 12-, 24-, 36-, 48-, 60 hr in cold exposure using rector thermometer (Kent Scientific, WD-20250–91) and rectal probe (Kent Scientific, RET-3). Before recording temperatures, measuring instruments were calibrated using ice-water bath each time. Probe was gently inserted into the mouse rectum at least 2 cm. Detailed data see **Supplementary file 4**.

## Intraperitoneal glucose tolerance test

Mice were fasted for 6 hr starting from 9 am to 3 pm on the testing day Glucose was administered intraperitoneally (2 g/kg body weight) and blood glucose levels were determined from tail vein blood samples using the ACCU-CHEK active glucometer at several time points post glucose injection.

## Cell implantation

Immortalized GFPpos and mBaSVF cells were differentiated under pro-adipogenic conditions, respectively, as described above. After 2-day induction, $1.3 \times 10^6$ of each group of committed cells were gently re-suspended and embedded in 110 µl Matrigel (Corning, 356231). The complex was subsequently injected subcutaneously into the left abdomen of C57BL/6 J wild-type mice at 8 weeks of age. To trace the fate and the locations of implanted cells, blue agarose beads (150–300 µm in diameter;

Bio-Rad, 153–7301) were included as an indicator in the implanted complexes. Mice were then kept under regular chow diet at ambient temperature, and ectopically formed fat pads were identified and confirmed by gross and histological examinations 2 week after injection. Accordingly, cell-implanted mice were subject to GTT at 2-week time point after cell implantation.

## Statistics

Statistical analyses were performed using GraphPad Prism 9.0 (GraphPad Software), and Excel (Microsoft). All data were represented as mean ± s.e.m, except where noted. A two-sample unpaired Student's t-test was used for two-group comparisons. One-way ANOVA followed by Tukey's test was used for multiple group comparisons, two-way repeated-measures ANOVA followed by Bonferroni's test was used for Seahorse measurement and GTT results from multiple groups. p values below 0.05 were considered significant throughout the study and is presented as $*p < 0.05$, $**p < 0.01$, or $*** < 0.001$.

## Acknowledgements

We thank members of the Chen Lab for providing technical advice and sharing reagents. We thank Dr. Konrad Basler of the University of Zurich for his kind gift of *Ctnnb1*[dm] mice. This work was supported by a Carol Lavin Bernick Faculty Grant from Tulane University, the John L and Mary Wright Ebaugh Endowed Chair Fund, and a grant (R01DK128907) from the NIH to YC. TY was supported by an American Heart Association Predoctoral Fellowship (20PRE35040002).

## Additional information

### Funding

| Funder | Grant reference number | Author |
|---|---|---|
| Tulane University | Carol Lavin Bernick Faculty Grant | YiPing Chen |
| Tulane University | the John L. and Mary Wright Ebaugh Endowed Chair Fund | YiPing Chen |
| National Institutes of Health | R01DK128907 | YiPing Chen |
| American Heart Association | 20PRE35040002 | Tianfang Yang |

The funders had no role in study design, data collection and interpretation, or the decision to submit the work for publication.

### Author contributions

Zhi Liu, Tian Chen, Conceptualization, Data curation, Formal analysis, Investigation, Methodology, Software, Visualization, Writing - original draft, Writing – review and editing; Sicheng Zhang, Investigation, S.Z. assisted with cell line establishment and collected data, Visualization; Tianfang Yang, Funding acquisition, Investigation, T.Y. helped to created transgenic mouse lines; Yun Gong, Formal analysis, Software, Y.G. helped to analyze bioinformatic data; Hong-Wen Deng, Formal analysis, H.D. helped to analyze bioinformatic data, Software; Ding Bai, Conceptualization, D.B. helped to design the initial experiments and provide mentoring and intellectual advice during the study, Supervision, Writing – review and editing; Weidong Tian, Conceptualization, Supervision, W.T. conceived the study, designed experiments, analyzed data, and edited the manuscript, Writing – review and editing; YiPing Chen, Conceptualization, Funding acquisition, Investigation, Supervision, Writing – review and editing

### Author ORCIDs

Zhi Liu http://orcid.org/0000-0001-6183-498X
YiPing Chen http://orcid.org/0000-0002-8628-7713

### Ethics

This study was performed in strict accordance with the recommendations in the Guide for the Care and Use of Laboratory Animals of the National Institutes of Health. All of the animals were handled according to approved institutional animal care and use committee (IACUC) protocols (#782) of Tulane University. All surgery was performed under sodium pentobarbital anesthesia, and every effort was made to minimize suffering.

### Decision letter and Author response

Decision letter https://doi.org/10.7554/eLife.77740.sa1
Author response https://doi.org/10.7554/eLife.77740.sa2

## Additional files

### Supplementary files

• Supplementary file 1. Primer sequences used for qRT-PCR.

• Supplementary file 2. Sheet 1: Enriched genes in scRNA-seq. Sheet 2: Overrepresented DNA motifs in scATAC-seq.

• Supplementary file 3. Pathway analyses results of DEGs enriched in iWAT-derived Wnt⁺ adipocytes in scRNA-seq.

• Supplementary file 4. Core body temperature results of T/L-DTA and control mice. Core body temperature below 34.5 °C was considered as hypothermia and such mice were subsequently euthanized.

• Supplementary file 5. Sheet 1: Significantly down-regulated genes of LF3-treated Wnt⁺ adipocytes in bulk RNA-seq. Sheet 2: KEGG pathway analysis results of DEGs down-regulated in LF3-treated Wnt⁺ adipocytes in bulk RNA-seq.

• Transparent reporting form

### Data availability

The GEO accession number for the scRNA-seq, bulk RNA-seq, and scATAC-seq data is GSE164747. Scripts used to process and analyze data in this paper have been deposited to GitHub: https://github.com/ychen-lab/Wnt-positive-adipocyte, (copy archived at swh:1:rev:9da40ea5cd877f8de0081d77722e7f74612113c8).

The following dataset was generated:

| Author(s) | Year | Dataset title | Dataset URL | Database and Identifier |
|---|---|---|---|---|
| Liu Z, Chen T, Tian W, Chen Y | 2022 | Discovery and functional assessment of a novel adipocyte population driven by intracellular Wnt/β-catenin signaling in mammals | https://www.ncbi.nlm.nih.gov/geo/query/acc.cgi?acc=GSE164747 | NCBI Gene Expression Omnibus, GSE164747 |

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
