## [Editor Report]

It is becoming increasingly clear that adipocytes are not homogenous, but rather comprise several distinct subtypes with specific physiologic functions. This work presents evidence for an unexpected subpopulation of adipocytes displaying atypical Wnt signaling. The data suggest a role of these adipocytes in thermogenesis, which could have importance for understanding energy homeostasis.

---

## [Decision Letter]

**Decision letter after peer review:**

Thank you for submitting your article "Discovery and functional assessment of a novel adipocyte population driven by intracellular Wnt/β-catenin signaling in mammals" for consideration by *eLife*. Your article has been reviewed by 3 peer reviewers, and the evaluation has been overseen by a Reviewing Editor and Mone Zaidi as the Senior Editor. The following individuals involved in the review of your submission have agreed to reveal their identity: Kosaku Shinoda (Reviewer #1); David Merrick (Reviewer #2).

Essential revisions:

1) The data presented for the beiging effect are insufficient to support the conclusions and would require major revision or could be simply removed as these findings do not significantly enhance the main conclusion that Wnt+ adipocytes exist.

2) It is necessary to address the issues raised by reviewer 2 regarding the apparent absence of Wnt+ non-adipocyte cells in the tissue. The existing single-nuclei RNA-Seq can be reanalyzed to determine if there are mature adipocytes in vivo with active Tcf/Lef signaling. This could eliminate the possibility that GFP signaling is a nonspecific signal from stromal cells, and would be necessary to validate the hypothesis that the fluorescent Wnt reporter is representative of physiologically relevant Tcf7l2 signaling.

3) It is desirable to strengthen the characterization of what Wnt signaling pathway is involved, and of its interaction with the insulin/PI3K/AKT/MTOR pathway, as outlined by Reviewer 3.

*Reviewer #1 (Recommendations for the authors):*

I found no major flaws in the method – this is a very impressive and innovative finding, worthy of publication in *eLife*. Below are suggestions for the authors to help strengthen the conclusion.

(1) The authors differentiated SVF into mature adipocytes before performing scRNA-seq. In general, differentiated adipocytes are prone to rupture and are not suitable for this method, thus authors should write more about how they successfully isolated Wnt+ and Wnt- cells using FACS prior to scRNA-esq.

(2) Related to (1). it is not clear from the methods section whether scRNA-seq library prep performed separately for GFP+ and GFP- cells and then co-clustered later bioinformatically? Or were GEMs produced with combined GFP+ and GFP-cells?

(3) What is the body composition and body weight curve of T/L-DTA and Wnt+-implanted under a high-fat diet? Did GTT improve independent of adiposity?

(4) Related to (3), food intake, energy expenditure, and fasting insulin should also be measured prior to the GTT.

(5) This is the most important point, since adipocytes are known to be highly autofluorescent, the littermate wild type of T/L-GFP should be observed with the same microscope setting and background GFP levels should be reported and included adjacent to Figure 1A.

*Reviewer #2 (Recommendations for the authors):*

Figure S1-1 E/F: shows a GFP pattern entirely consistent with stromal cells surrounding an unlabeled mature adipocyte: multiple nuclei, irregular/incomplete border thickness, strongest GFP staining in spindle-shaped fibroblast cells between adipocytes.

Figure 2 C: Endothelial cells, some leukocytes and some fibroblasts engage very active Wnt signaling and are Adipoq-negative, thus these cell types should retain high nuclear GFP staining in the Adipoq;Ctnnb1 mouse model. These cells types make up a large portion of the total nuclei in adipose tissues. Thus is it unclear why there are no GFP+ nuclei between the adipocytes (i.e. it appears that almost all GFP+ labeled has disappeared in this mouse model).

Figure 2E/F: The authors should attempt to provide some explanation as to why LF3 causes such a dramatic inhibition of differentiation in the Wnt-negative adipocytes. If the inhibitor is specific to disrupting Tcf7l1-β-catenin interaction, why does it have such a profound effect on these Wnt-negative cells in which that interaction should not be occurring?

Figure 4 E/G, 5S^-1^: It is not possible (especially in the iBAT/beige) to confidently assign the GFP+ nuclei to adipocytes based on the images presented. The only conclusion2`````` that can be drawn from these experiments is that Wnt+ cells (both stromal and adipocyte) are decreased with the interventions.

Figure 5 A/B/D: The proximity data are not convincing given the presence of GFP+ nuclei within regions of high membrane potential (to the left and below the yellow box), and in regions without beiging. Thus the conclusions that Wnt+ cells (again they cannot be confidently identified as adipocyte nuclei) regulate beiging is not supported by these data.

Figure 5 E: Cold exposure and tamoxifen administration are both well-documented stimuli for de novo adipogenesis from precursor cells.

Figure S1-1 E shows robust labeling of Wnt+ progenitor cells intercalated between the mature adipocytes in iWAT. Thus is it plausible that the lineage-traced beige adipocytes are derived from Wnt+ precursor cells? The authors' conclusion that the T/L-CreER lineage tracing shows "unambiguous evidence" that these adipocytes are derived from Wnt+ mature adipocytes is overstated and should be removed.

Figure S5-1: Fabp4 is expressed in adipocyte progenitor cells, which are also Wnt+, thus these cell populations will likely be depleted with diphtheria toxin injection and could account for the loss of GFP+ nuclei in the tissues. Depletion of these progenitors could impair de-novo beige adipogenesis leading to the observed phenotype.

*Reviewer #3 (Recommendations for the authors):*

– The involvement of TCF7L2 is also probed using a specific inhibitor of the β-catenin/TCF7L2 interactions, LF3, which inhibited reporter expression. Inhibition of canonical Wnt signaling was without effect. The use of additional activators and inhibitors of the destruction pathway would broaden the comparability of this study to previous literature on the role of Wnt in adipose differentiation.

– It is not clear why scRNASeq was chosen over RNASeq on the population, since the fat content of adipocytes may preclude full characterization of the most differentiated cells. RNASeq of a set of independently derived Wnt+ and Wnt- adipocytes may be more informative.

– An additional concern is that diphtheria toxin-induced cell death will lead to tissue inflammation, with potential functional effects on thermogenesis. The degree of cell death and inflammation should be measured and reported.

– The impaired beiging in the Akita mouse should be strengthened by RT-PCR and western blotting for UCP1 in the entire fat pads of a cohort of independent control and Akita mice. These results, as well as measurements of whether the mice are able to defend their body temperature in response to a cold challenge, would be required to unambiguously conclude that Akita mice display impaired beiging.

– Implantation experiments need measurement of the volume of the tissues formed from implanted cells and some basic histological metrics including the degree of vascularization.

---

## [Author Response]

Essential revisions:1) The data presented for the beiging effect are insufficient to support the conclusions and would require major revision or could be simply removed as these findings do not significantly enhance the main conclusion that Wnt+ adipocytes exist.

We believe that the effect of Wnt^+^ adipocytes on beiging response is a critical function of such adipocytes and an important part of this paper, we thus would like to keep the results in the paper. To support this conclusion, we now present additional evidence with two different mouse lines that exhibit similarly compromised beiging response upon cold challenge: 1) to avoid potential side effects of tamoxifen in triggering low level of adipocyte apoptosis and inducing non-physiological beiging, we created a Tcf/Lef-rtTA mouse line and used it to ablate Wnt^+^ adipocytes upon compounding with TRE-Cre and Fabp4-Flex-DTA mice; 2) to avoid potential side effects of DTA-induced cell death on adipose tissues, we compounded the Tcf/Lef-rtTA allele with TRE-Cre and floxed *Pparg* alleles (*Pparg*^F/F^) to prevent the differentiation of Wnt^+^ adipocytes. These new results are included in the revision as supplemental results (Figure 5—figure supplement 2).

2) It is necessary to address the issues raised by reviewer 2 regarding the apparent absence of Wnt+ non-adipocyte cells in the tissue. The existing single-nuclei RNA-Seq can be reanalyzed to determine if there are mature adipocytes in vivo with active Tcf/Lef signaling. This could eliminate the possibility that GFP signaling is a nonspecific signal from stromal cells, and would be necessary to validate the hypothesis that the fluorescent Wnt reporter is representative of physiologically relevant Tcf7l2 signaling.

We have never seen any GFP-positive cells in freshly isolated SVFs by FACS in many experiments. A piece of representative FACS results is now included in the revision (Figure 2—figure supplement 1A). GFP expression was also never seen in plated SVFs (Figure 2A). We therefore believe that although some stromal cells may possess basal Wnt signaling activity, the lack of Wnt^+^ non-adipocyte cells in adipose tissues could be attributed to the relatively low level of canonical Wnt signaling activity, which is below the threshold for activating Tcf/Lef-GFP reporter. This assumption is supported by 1) while Wnt^-^ adipocytes do not express the Tcf/Lef-GFP reporter, LF3 administration still delays maturation of Wnt^-^ adipocytes, suggesting that Wnt^-^ adipocytes have functional basal Wnt signaling activity (Figure 2—figure supplement 1E). This conclusion is in line with several previous reports that showed the presence of basal Wnt signaling activity in overall mature adipocytes and its role in controlling de novo lipogenesis (Bagchi et al., 2020; Geoghegan et al., 2019; Mori et al., 2012); 2) shown in Figure 1—figure supplement 3B, human stromal cells infected with Tcf/Lef-GFP reporter exhibited GFP expression in almost all cells under pro-osteogenic induction that stimulates hyperactivation of Wnt canonical signaling. In addition, other well-known Wnt^+^ cells in the T/L-GFP mice we used in the current studies such as muscle cells and dermal cells express GFP signals specifically, validating the specificity of the reporter allele.

We took the advice by Reviewer 2 and intersected our scRNA-seq data on Wnt^+^ adipocytes with the published single-nucleus sequencing (sNuc-seq) dataset of mouse iWAT (Emont et al., 2022). Because the activation of Tcf/Lef signaling in the Wnt^+^ adipocytes is relied on AKT/mTOR signaling but not the conventional Wnt ligands and receptors, those traditional downstream markers of Wnt signaling such *Axins* were not found specifically enriched in Wnt^+^ adipocytes. Therefore, the AKT/mTOR-dependent Wnt signaling in Wnt^+^ adipocytes appears to regulate the expression of genes distinct from that controlled by the conventional Wnt signaling pathway. This conclusion is supported by our recent studies that inhibition of this AKT/mTOR-dependent Wnt signaling by LF3 in Wnt^+^ adipocytes negatively impact pathways implicated in “PI3K/Akt signaling”, “insulin signaling”, “thermogenesis”, and “fatty acid metabolism” et al. (see below for details). However, we found that one cluster (mAd3) of sNuc-seq dataset, which is relatively enriched in *Tcf7l2*, expresses remarkably high levels of *Cyp2e1* as well as *Cfd* that encodes Adipsin. These genes, regarded as hallmark of mAd3 cluster, are also uniquely or highly expressed in Wnt^+^ adipocytes. Interestingly, the percentage of mAd3 among the total iWAT adipocytes in chow-fed male group is about 5%, which is very close to that of Wnt^+^ adipocytes in vivo (~7%). Thus, mAd3 possibly represents Wnt^+^ adipocytes in iWAT. These analyses are included in the revision.

3) It is desirable to strengthen the characterization of what Wnt signaling pathway is involved, and of its interaction with the insulin/PI3K/AKT/MTOR pathway, as outlined by Reviewer 3.

To explore the downstream targets of the intracellular Wnt signaling in Wnt^+^ adipocytes, we have performed RNA-seq on Wnt^+^ adipocytes with and without LF3 treatment. The KEGG pathway analysis of DEGs (fold change > 2, FDR < 0.05) showed that the primarily affected (downregulated) pathways in LF3-treated Wnt^+^ adipocytes are “PI3K/Akt signaling”, “thermogenesis”, “insulin signaling”, “fatty acid metabolism” et al. These results indicate that the intracellular Wnt signaling indeed mediates the function of the insulin/PI3K/AKT/mTOR pathway. The results are now included in the revision (Figure 4—figure supplement 1F, G).

Reviewer #1 (Recommendations for the authors):I found no major flaws in the method – this is a very impressive and innovative finding, worthy of publication in eLife. Below are suggestions for the authors to help strengthen the conclusion.(1) The authors differentiated SVF into mature adipocytes before performing scRNA-seq. In general, differentiated adipocytes are prone to rupture and are not suitable for this method, thus authors should write more about how they successfully isolated Wnt+ and Wnt- cells using FACS prior to scRNA-esq.

Since GFP expression from the T/L-GFP allele in Wnt^+^ adipocytes is activated at early stage of adipogenic differentiation with relatively less lipid accumulation, this allows successful isolation by FACS. We have made it clear in the revision.

(2) Related to (1). it is not clear from the methods section whether scRNA-seq library prep performed separately for GFP+ and GFP- cells and then co-clustered later bioinformatically? Or were GEMs produced with combined GFP+ and GFP-cells?

Yes, we performed separate library preparation and sequenced the libraries together. We now make this point clear in the revision.

(3) What is the body composition and body weight curve of T/L-DTA and Wnt+-implanted under a high-fat diet? Did GTT improve independent of adiposity?

We did not perform these studies but agree that they warrant future studies.

(4) Related to (3), food intake, energy expenditure, and fasting insulin should also be measured prior to the GTT.

Since our lab studies developmental biology and lacks essential equipment such as MRI and metabolic cages for these studies, we are not able to do these experiments at this moment. Hopefully these suggested studies will be addressed in the near future through collaborations.

(5) This is the most important point, since adipocytes are known to be highly autofluorescent, the littermate wild type of T/L-GFP should be observed with the same microscope setting and background GFP levels should be reported and included adjacent to Figure 1A.

Figure 2C serves as a control for such purpose.

Reviewer #2 (Recommendations for the authors):Figure S1-1 E/F: shows a GFP pattern entirely consistent with stromal cells surrounding an unlabeled mature adipocyte: multiple nuclei, irregular/incomplete border thickness, strongest GFP staining in spindle-shaped fibroblast cells between adipocytes.

These GFP-labeled dots and irregular structures are most likely fragmented GFP-labeled plasma membrane and some acellular structures with autofluorescence, likely due to tamoxifen administration. As shown in Figure 5—figure supplement 2D in which TRECre;Tcf/Lef-rtTA;*Rosa26R*^mTmG^ mice exhibit much reduced background. In addition, as shown in the newly added result (Figure 2—figure supplement 1A), GFP-positive cells were never seen in freshly isolated SVFs of T/L-GFP iWAT determined by FACS.

Figure 2 C: Endothelial cells, some leukocytes and some fibroblasts engage very active Wnt signaling and are Adipoq-negative, thus these cell types should retain high nuclear GFP staining in the Adipoq;Ctnnb1 mouse model. These cells types make up a large portion of the total nuclei in adipose tissues. Thus is it unclear why there are no GFP+ nuclei between the adipocytes (i.e. it appears that almost all GFP+ labeled has disappeared in this mouse model).

We believe that although these non-adipose cells possess basal Wnt signaling activity, the lack of Wnt^+^ non-adipocyte cells in adipose tissues could be attributed to the relatively low level of canonical Wnt signaling activity, which is below the threshold to activate Tcf/Lef-GFP reporter. This assumption is supported by 1) while Wnt^-^ adipocytes do not express Tcf/Lef-GFP reporter, LF3 administration still delays maturation of Wnt^-^ adipocytes, suggesting that Wnt^-^ adipocytes have functional basal Wnt signaling activity (Figure 2—figure supplement 1E). This conclusion is in line with several previous reports that showed the presence of basal Wnt signaling activity in overall mature adipocytes and its role in controlling de novo lipogenesis (Bagchi *et al.*, 2020; Geoghegan *et al.*, 2019; Mori *et al.*, 2012); 2) shown in Figure 1—figure supplement 3B, human stromal cells infected with Tcf/Lef-GFP reporter exhibited GFP expression in almost all cells under pro-osteogenic induction that stimulates hyperactivation of canonical Wnt signaling. In addition, other well-known Wnt^+^ cells in the T/L-GFP mice we used in the current studies such as muscle cells (Figure 2C) and dermal cells (Figure 5—figure supplement 2C) express GFP signals specifically, validating the specificity of the reporter allele.

Figure 2E/F: The authors should attempt to provide some explanation as to why LF3 causes such a dramatic inhibition of differentiation in the Wnt-negative adipocytes. If the inhibitor is specific to disrupting Tcf7l1-β-catenin interaction, why does it have such a profound effect on these Wnt-negative cells in which that interaction should not be occurring?

Emerging evidence suggests that basal activity of canonical Wnt signaling is present and plays functions in mature adipocytes, particularly is required for de novo lipogenesis of adipocytes (Bagchi *et al.*, 2020). Since LF3 could disrupt Tcf7l2-β-catenin interaction in both conventional Wnt canonical signaling in Wnt^-^ adipocytes and the AKT/mTOR-dependent intracellular Wnt signaling in Wnt^+^ adipocytes, the observation that LF3 treatment delayed Wnt^-^ adipocyte maturation is in line with the previous publication (Bagchi *et al.*, 2020).

Figure 4 E/G, 5S^-1^: It is not possible (especially in the iBAT/beige) to confidently assign the GFP+ nuclei to adipocytes based on the images presented. The only conclusion2`````` that can be drawn from these experiments is that Wnt+ cells (both stromal and adipocyte) are decreased with the interventions.

As we explained above, GFP^+^ cells were never found in freshly isolated SVFs in the T/L-GFP reporter mouse line that was used in the current studies.

Figure 5 A/B/D: The proximity data are not convincing given the presence of GFP+ nuclei within regions of high membrane potential (to the left and below the yellow box), and in regions without beiging. Thus the conclusions that Wnt+ cells (again they cannot be confidently identified as adipocyte nuclei) regulate beiging is not supported by these data.

The results shown in Figure 5A/B provided the initial hint for the implication of Wnt^+^ adipocytes in regulation of beiging by paracrine effect. Figure 5D showed close physical association of beige adipocytes with Wnt^+^ adipocytes after 4-day cold challenge. The proximity of Wnt^+^ and beige adipocytes is better exemplified in initial beiging response (after 2-day cold challenge), as shown in Figure 5—figure supplement 1C. Actually, the real supporting results came from the in vivo cold challenge experiments on T/L-DTA mice that exhibited severely impaired beiging response. Our new results using TRE-Cre;Tcf/Lef-rtTA;Fabp4-Flex-DTA mice as well as TRE-Cre;Tcf/Lef-rtTA;*Pparg*^F/F^ mice showed similar compromised beiging response upon cold challenge.

Figure 5 E: Cold exposure and tamoxifen administration are both well-documented stimuli for de novo adipogenesis from precursor cells.

We agree with this reviewer. To avoid potential side effects of tamoxifen, we created a Tcf/Lef-rtTA mouse line and used it to ablate Wnt^+^ adipocytes upon compounding with TRE-Cre and Fabp4-Flex-DTA mice. These results are included in the revision.

Figure S1-1 E shows robust labeling of Wnt+ progenitor cells intercalated between the mature adipocytes in iWAT. Thus is it plausible that the lineage-traced beige adipocytes are derived from Wnt+ precursor cells? The authors' conclusion that the T/L-CreER lineage tracing shows "unambiguous evidence" that these adipocytes are derived from Wnt+ mature adipocytes is overstated and should be removed.

As mentioned above, GFP reporter is not activated in freshly isolated SVFs including Wnt^+^ precursor cells. This point is further supported by the evidence that the percentage of Wnt^+^ adipocytes did not increase after cold and CL316,243 stimulations (Figure 5-figurte supplement 1E). In addition, the T/L-GFP reporter is never activated until adipogenic differentiation in the immortalized precursor cell lines (GFPpos^-1^, and -2) of Wnt^+^ adipocytes.

Figure S5-1: Fabp4 is expressed in adipocyte progenitor cells, which are also Wnt+, thus these cell populations will likely be depleted with diphtheria toxin injection and could account for the loss of GFP+ nuclei in the tissues. Depletion of these progenitors could impair de-novo beige adipogenesis leading to the observed phenotype.

As above discussed, the T/L-GFP allele is not activated in SVFs, thus it would not cause ablation of precursor cells of Wnt^+^ adipocytes in T/L-DTA mice.

Reviewer #3 (Recommendations for the authors):– The involvement of TCF7L2 is also probed using a specific inhibitor of the β-catenin/TCF7L2 interactions, LF3, which inhibited reporter expression. Inhibition of canonical Wnt signaling was without effect. The use of additional activators and inhibitors of the destruction pathway would broaden the comparability of this study to previous literature on the role of Wnt in adipose differentiation.

Thank you for the suggestions.

– It is not clear why scRNASeq was chosen over RNASeq on the population, since the fat content of adipocytes may preclude full characterization of the most differentiated cells. RNASeq of a set of independently derived Wnt+ and Wnt- adipocytes may be more informative.

With scRNA-seq, it would be more convincing to identify specific subpopulation of cells, as adipocytes are well known to be heterogenous. In addition, the cell lines are immortalized and biasedly selected. Thus, they may not best reflect the overall true gene expression signature.

– An additional concern is that diphtheria toxin-induced cell death will lead to tissue inflammation, with potential functional effects on thermogenesis. The degree of cell death and inflammation should be measured and reported.

To avoid potential side effects of DTA-induced cell death on adipose tissues, we compounded the Tcf/Lef-rtTA allele with TRE-Cre and floxed *Pparg* alleles (*Pparg*^F/F^) to prevent the differentiation of Wnt^+^ adipocytes. These new results are included in the revision as supplemental results (Figure 5—figure supplement 2G).

– The impaired beiging in the Akita mouse should be strengthened by RT-PCR and western blotting for UCP1 in the entire fat pads of a cohort of independent control and Akita mice. These results, as well as measurements of whether the mice are able to defend their body temperature in response to a cold challenge, would be required to unambiguously conclude that Akita mice display impaired beiging.

Yes, RT-PCR and Western blotting results for UCP1 expression are included.

– Implantation experiments need measurement of the volume of the tissues formed from implanted cells and some basic histological metrics including the degree of vascularization.

We used the same number (1.3 x 10^6^) of both control and Wnt^+^ adipocytes for each implantation in our studies. However, the size of fat pads varied due to different compaction and distribution, which makes it very difficult to measure the degree of vascularization.

Reference

Bagchi, D.P., Li, Z., Corsa, C.A., Hardij, J., Mori, H., Learman, B.S., Lewis, K.T., Schill, R.L., Romanelli, S.M., and MacDougald, O.A. (2020). Wntless regulates lipogenic gene expression in adipocytes and protects against diet-induced metabolic dysfunction. Mol Metab *39*, 100992. 10.1016/j.molmet.2020.100992.

Emont, M.P., Jacobs, C., Essene, A.L., Pant, D., Tenen, D., Colleluori, G., Di Vincenzo, A., Jorgensen, A.M., Dashti, H., Stefek, A., et al. (2022). A single-cell atlas of human and mouse white adipose tissue. Nature *603*, 926-933. 10.1038/s41586-022-04518-2.

Geoghegan, G., Simcox, J., Seldin, M.M., Parnell, T.J., Stubben, C., Just, S., Begaye, L., Lusis, A.J., and Villanueva, C.J. (2019). Targeted deletion of Tcf7l2 in adipocytes promotes adipocyte hypertrophy and impaired glucose metabolism. Mol Metab *24*, 44-63. 10.1016/j.molmet.2019.03.003.

Mori, H., Prestwich, T.C., Reid, M.A., Longo, K.A., Gerin, I., Cawthorn, W.P., Susulic, V.S., Krishnan, V., Greenfield, A., and Macdougald, O.A. (2012). Secreted frizzled-related protein 5 suppresses adipocyte mitochondrial metabolism through WNT inhibition. J Clin Invest *122*, 2405-2416. 10.1172/JCI63604.

Xu, P., Li, J., Liu, J., Wang, J., Wu, Z., Zhang, X., and Zhai, Y. (2017). Mature adipocytes observed to undergo reproliferation and polyploidy. FEBS Open Bio *7*, 652-658. 10.1002/2211-5463.12207.